# The genetic heterogeneity and drug resistance mechanisms of relapsed refractory multiple myeloma

Josh N. Vo[1,2,31], Yi-Mi Wu[1,3,31], Jeanmarie Mishler[1], Sarah Hall[1], Rahul Mannan [1,3], Lisha Wang[1], Yu Ning[1], Jin Zhou[1], Alexander C. Hopkins[1], James C. Estill[1], Wallace K. B. Chan [4], Jennifer Yesil[5], Xuhong Cao[1,3,6], Arvind Rao[2,7,8,9], Alexander Tsodikov[10], Moshe Talpaz[11,12], Craig E. Cole[13], Jing C. Ye[11,12], Multiple Myeloma Research Consortium*, P. Leif Bergsagel [14], Daniel Auclair[5], Hearn Jay Cho[5], Dan R. Robinson [1,3,32✉] & Arul M. Chinnaiyan [1,3,6,12,15,32✉]

Multiple myeloma is the second most common hematological malignancy. Despite significant advances in treatment, relapse is common and carries a poor prognosis. Thus, it is critical to elucidate the genetic factors contributing to disease progression and drug resistance. Here, we carry out integrative clinical sequencing of 511 relapsed, refractory multiple myeloma (RRMM) patients to define the disease's molecular alterations landscape. The NF-κB and RAS/MAPK pathways are more commonly altered than previously reported, with a prevalence of 45–65% each. In the RAS/MAPK pathway, there is a long tail of variants associated with the RASopathies. By comparing our RRMM cases with untreated patients, we identify a diverse set of alterations conferring resistance to three main classes of targeted therapy in 22% of our cohort. Activating mutations in *IL6ST* are also enriched in RRMM. Taken together, our study serves as a resource for future investigations of RRMM biology and potentially informs clinical management.

[1] Michigan Center for Translational Pathology, University of Michigan, Ann Arbor, MI, USA. [2] Department of Computational Medicine and Bioinformatics, University of Michigan, Ann Arbor, MI, USA. [3] Department of Pathology, University of Michigan, Ann Arbor, MI, USA. [4] Department of Pharmacology, University of Michigan, Ann Arbor, MI, USA. [5] Multiple Myeloma Research Foundation, Norwalk, CT, USA. [6] Howard Hughes Medical Institute, University of Michigan, Ann Arbor, MI, USA. [7] Department of Biostatistics, University of Michigan, Ann Arbor, MI, USA. [8] Department of Radiation Oncology, University of Michigan, Ann Arbor, MI, USA. [9] Department of Biomedical Engineering, University of Michigan, Ann Arbor, MI, USA. [10] Department of Biostatistics, School of Public Health, University of Michigan, Ann Arbor, MI, USA. [11] Department of Internal Medicine, Division of Hematology and Oncology, University of Michigan, Ann Arbor, MI, USA. [12] Rogel Cancer Center, University of Michigan, Ann Arbor, MI, USA. [13] College of Human Medicine, Breslin Cancer Center, Michigan State University, East Lansing, MI, USA. [14] Division of Hematology, Department of Internal Medicine, Mayo Clinic, Phoenix, AZ, USA. [15] Department of Urology, University of Michigan, Ann Arbor, MI, USA. [31] These authors contributed equally: Josh N. Vo, Yi-Mi Wu. [32] These authors jointly supervised this work: Dan R. Robinson, Arul M. Chinnaiyan. *A list of authors and their affiliations appears at the end of the paper. ✉email: danrobi@umich.edu; arul@umich.edu

Multiple myeloma (MM) is a malignancy of plasma cells, mature B lymphocytes dedicated to producing immunoglobulins. The median age of diagnosis of MM is 70 years, and MM affects individuals of African descent two- to three-fold more than those of European descent, accounting for significant health disparities[1,2]. Over the last two decades, many novel treatments have been developed for MM, including proteasome inhibitors, immunomodulatory drugs, monoclonal antibodies, and CAR-T-cell therapies[3]. The 5-year relative survival rate across all stages combined is 54%, which has improved based on the advent of these new treatments[4]. Despite this, the majority of MM patients suffer a relapse, and each subsequent relapse limits treatment options and reduces the ability to control disease progression[1,5]. Thus, understanding the genetic heterogeneity of relapsed, refractory MM (RRMM) will shed light on MM disease progression as well as elucidate therapeutic resistance mechanisms. Newly diagnosed MM (NDMM), like other primary cancers, has been genomically dissected by a number of groups, including the Multiple Myeloma Research Foundation's (MMRF) CoMMpass Study (NCT01454297)[6]. Less is known about the genomic heterogeneity and resistance mechanisms of RRMM.

In this work, as part of the MMRF's molecular profiling initiative (NCT02884102) which included 22 academic medical centers, we carry out clinical-grade targeted sequencing (tumor/normal) and whole transcriptome sequencing of 511 RRMM patients and reanalyzed equivalent data from 965 patients enrolled in the CoMMpass Study to systematically compare alterations in RRMM with NDMM. Our uniform integrative analyses uncover a wide range of genetic alterations, implicate known oncogenic MM pathways often at a much higher prevalence than previously known, and provide a comprehensive genetic basis for drug resistance mechanisms in RRMM.

## Results

### The landscape of genomic alterations in relapsed refractory multiple myeloma.

As part of the MMRF molecular profiling initiative, a consecutive series of 762 RRMM patients were enrolled and provided CLIA genomic sequencing from May 2017 to June 2020. As part of this precision oncology study, molecular reports were provided to the participating physicians within a 10-day turnaround period. After the exclusion of patients that either had allogeneic stem cell transplants (allo-SCT), or had low tumor purity (<20%) following CD138 + selection, we compiled a cohort of 511 RRMM patients with comprehensive 1700-gene tumor/normal DNA sequencing and whole transcriptome sequencing[7–10] for integrative analyses (Supplementary Fig. 1a and Supplementary Data 1). Overall, RRMM samples were contributed by 22 academic medical centers that participate in the Multiple Myeloma Research Consortium (MMRC), with the University of Michigan Rogel Cancer Center, Mt. Sinai Medical Center, and Hackensack Medical Center as the top three enrolling centers (Supplementary Fig. 1b). The cohort was comprised of 53.3% males and 46.7% females. Light chain type statistics included 52.4% kappa, 31.5% lambda, and 0.4% biclonal; 15.7% did not have available information. After CD138 + selection, the average tumor content of our RRMM cohort was 62%. For each patient, we performed targeted sequencing (Onco1700 panel)[10] on DNA from tumor and matched normal samples to call somatic and germline genetic aberrations (Methods, Supplementary Data 2). Mean target coverages for tumor and normal libraries were 655X and 483X, respectively, which is a depth conducive for subclonal assessments. RNA-sequencing for all but one tumor sample was available and performed by capture transcriptome sequencing with a cohort average of 44.5 M uniquely mapped reads. Our computational pipeline[9] also called

copy-number alterations (CNAs) (Supplementary Data 3), gene fusions (Supplementary Data 4), and provided gene expression levels on a per-sample basis (Supplementary Fig. 1c). FASTQ files of 1108 NDMM samples from the CoMMpass Study were downloaded from dbGaP as of August 2020 and analyzed with the same bioinformatics pipeline to facilitate comparisons with RRMM. In total, 965 complete CoMMpass cases were available for downstream integrative analyses (Methods and Supplementary Fig. 1a, c).

Employing an ensemble approach of statistical tools[11] to discover cancer drivers based on single nucleotide variants (SNV) and indels, we identified a set of 43 high-confident, significantly mutated genes (Fig. 1a and Supplementary Data 2). Their biological functions can be classified into the following categories: (1) RAS-MAPK pathway (KRAS, NRAS, BRAF, PTPN11, NF1, IL6ST), (2) NF-κB pathway (CYLD, TRAF3, TRAF2, NFKBIA, IRAK1), (3) MYC pathway (MYC, MAX, EP300, CREBBP, SP3), (4) cell cycle and DNA damage checkpoints (TP53, RB1, CDKN2C, CDKN1B, ATM, FGFR3, LATS2), (5) RNA processing machinery (DIS3, FAM46C, DDX3X, DDX5), (6) epigenetic modifiers and transcriptional co-activators/co-repressors (KDM3B, SETD2, ARID1A, ARID2, MBD1, IDH1, BCORL1, ATAD2B), (7) B lymphocyte development (PRDM1, SP140, UBR5, IRF4), and (8) genes that likely acquired mutations due to MM treatment (CRBN, CUL4B, NR3C1) (Supplementary Fig. 2a, b)[12]. The average mutation rate of RRMM was 3.43 point mutations/megabase. Hypermutation was observed in 9.8% of cases due to the APOBEC mutational process and in another 2.7% due to undefined mechanisms (Supplementary Fig. 2c). Mutations in cancer driver genes were also detected in matched-normal sequencing libraries with a wide range of variant allelic fraction (VAF) due to the presence of circulating tumor cells (CTCs) in advanced patients (Supplementary Fig. 2d).

Global copy-number analyses identified recurrent arm-level and chromosome-level gain of 1q, 3, 5, 7, 9, 11, 15, 17q, 19, and 21 (e.g., "hyperdiploid") and loss of 6q, 8p, 13,16, 22q and X (Fig. 1b, Supplementary Fig. 3a, and Supplementary Data 3). Frequent focal losses tended to center at or near known tumor suppressors in MM, such as GFI1, FAM46C, CDKN2C, ARID1B, NKX3-1, CDKN2A/B, BIRC2/3, CDKN1B, RB1, TRAF3, and CYLD, while focal gains were at or near oncogenes including MYC, CCND1, and TXNDC5 (Fig. 1b)[13]. Interestingly, there were recurrent homozygous deletions of diaphanous-related formin 2 (DIAPH2) on the X chromosome. These deletions were focal, affecting single exons or a group of exons, and affected both males and females in NDMM and RRMM (Supplementary Fig. 3b, c).

Integrative analysis of trinucleotide mutational signatures, gene expression, and copy-number identified distinct transcriptional signatures associated with high expression (presumably due to translocation) of WHSC1, CCND1, and MAF family genes (MAF, MAFA, and MAFB) (Supplementary Fig. 4a, b). As previously reported, samples with high expression of MAF family genes tended to associate with the APOBEC-enriched trinucleotide mutational signature (Supplementary Fig. 4a)[14] and had elevated expression of APOBEC3G (Supplementary Fig. 4c). A small subset (2.9%) of patients with high expression of CCND1 also exhibited high expression of pre-B cell markers, such as FCER2 (CD23), VPREB3, PAX5, and TNFRSF13C (Supplementary Fig. 4a, d), a finding which was also observed in NDMM[15].

### Highly prevalent, diverse mechanisms of NF-κB pathway activation.

The NF-κB pathway functions as an anti-apoptotic signal in myeloma cells and thus, mutations that lead to constitutive activation of NF-κB are selected for[16–18]. Our findings in

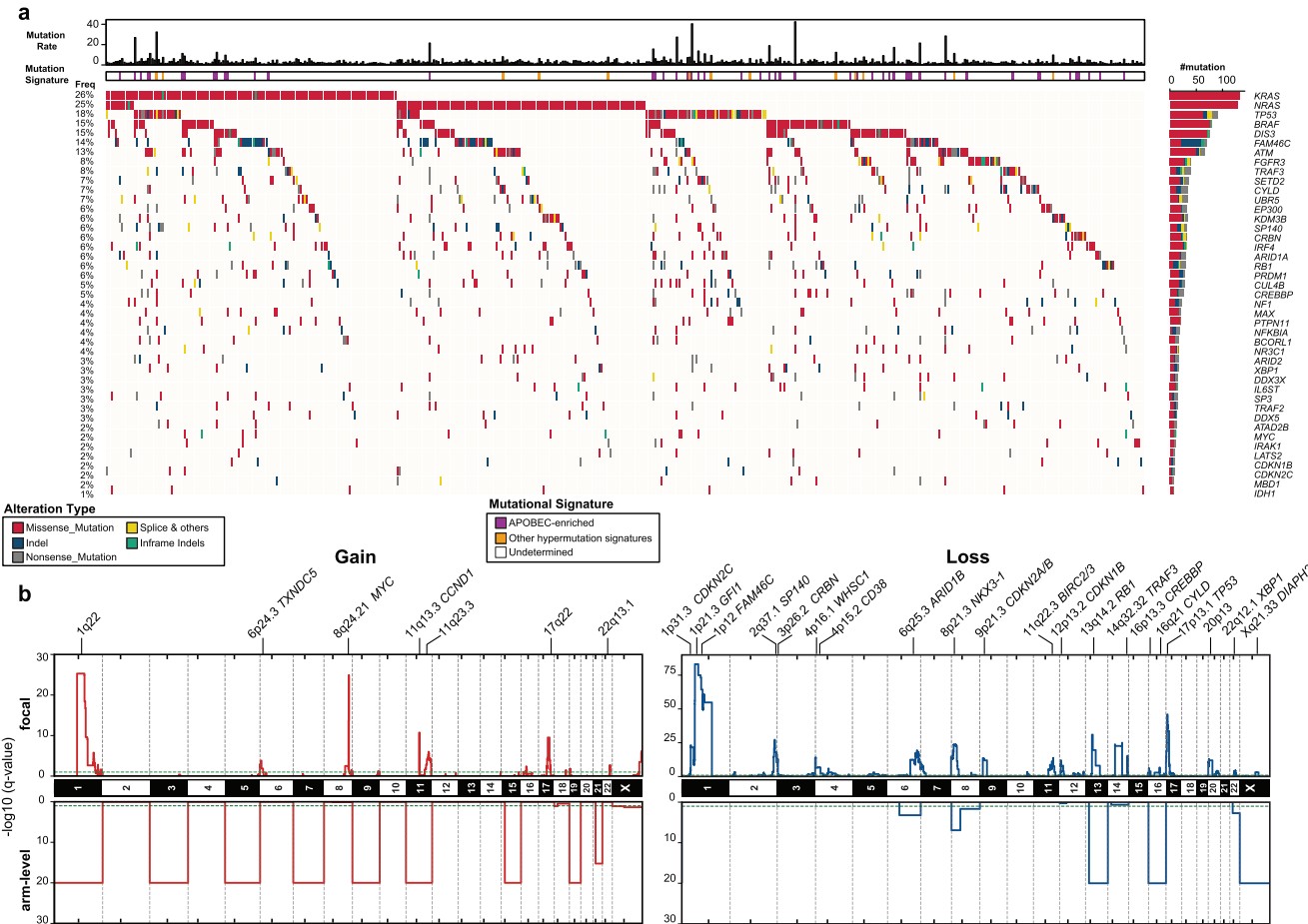

**Fig. 1 The landscape of somatic alterations in relapsed refractory multiple myeloma. a** Oncoprint of point mutations and small indels for significantly mutated genes in 511 cases of relapsed and refractory multiple myeloma (RRMM) from the MMRF Molecular Profiling Initiative. The panel of significant genes was derived from an ensemble approach for statistical testing (Methods). Classes of alterations are indicated in the legend. The upper histogram represents the mutation rate (number of point mutations per megabase) per case. APOBEC and non-APOBEC hypermutated cases are indicated. The right histogram represents the number of patients with an indicated mutation in the cohort, with percentages provided on the left. **b** Consensus plot for arm-level (lower panel) and focal copy-number alterations (upper panel). Gains are shown in red and losses in blue. Relevant genes in the recurrent gain or loss regions are indicated. The q-values were calculated by GISTIC2.0. Horizontal dashed green lines correspond to q value = 0.05.

RRMM were in line with previous studies[17] in NDMM in that genes involved in alternative (non-canonical) NF-κB signaling via the cell surface TNF family receptors, including *CD40*, *LTBR*, *TNFRSF17* (*BCMA*), and *TNFRSF13B* (*TACI*), were the most frequently affected by alterations (Fig. 2a, b). These alterations also induced robust activation of a previously defined NF-κB transcriptomic signature[16] (Fig. 2a, top). Our integrative cross-cohort analysis further identified in-frame insertions and deletions in the transmembrane domains of *TNFRSF17* (NDMM n = 7, RRMM n = 4) and *CD40* (NDMM n = 3) (Fig. 2c, d). Cloning of these mutations confirmed their activating potential in a cell-based, NF-κB reporter assay (Supplementary Fig. 5a, b). Thus, we hypothesize that in-frame indels involving amino acids in the transmembrane domain of these cell surface receptors induce local conformational alterations that facilitate ligand-independent oligomerization, leading to constitutive activation of downstream signaling. In addition to alterations in the alternative NF-κB pathway, our integrative analysis revealed recurrent alterations in genes of the classical (canonical) NF-κB signaling pathway, including Toll-like receptors (TLR), B cell receptor (BCR), and TNF-α (Fig. 2a, b right).

*MAP3K14* (*NIK*), the central kinase of the alternative NF-κB pathway, was also frequently truncated at the N-terminus by rearrangements that form in-frame fusion transcripts (Supplementary

Fig. 5c)[19]. These chimeric products have an intact kinase domain but lack the TRAF3 binding domain, and thus escape cIAP-TRAF2-TRAF3-mediated proteasomal degradation and remain stabilized in MM cells[20]. We also observed N-terminal intragenic deletions of *MAPK314* in both cohorts (Fig. 2e) which result in in-frame transcripts with a translation start site (methionine) located before the kinase domain. More complex rearrangements were out of frame or apparently lacked a methionine before the kinase domain. These co-occurred with a secondary event to restore the translation frame or introduce a de novo methionine. Examples of such a "second hit" were a frameshift mutation (P254fs) or intron retention/de novo splice site (Fig. 2e and Supplementary Fig. 5d). Interestingly, one case harbored a start-loss (M1I) mutation of *MAP3K14* while maintaining a robust NF-κB transcriptomic signature (Fig. 2e, last row). MAP3K14 was recently reported to have an N-terminal binding site for BIRC2 (c-IAP1)[21]. The start-loss mutation at M1 may force an alternative translation site (M4), thus disrupting the binding of BIRC2 and evading proteolytic degradation by the cIAP-TRAF2-TRAF3 complex. In-frame C-terminal fusions and deletions were also observed in *NFKB2* (p100) and *NFKB1* (p105) (Fig. 2f and Supplementary Fig. 6a). These rearrangements had breakpoints located in the ankyrin repeats, the domains on the precursor forms (p100 and p105) that bind the preformed NF-κB dimers (e.g., p50:RelA and p52:RelB), and thus disrupt the inhibitory activities of the precursors[22].

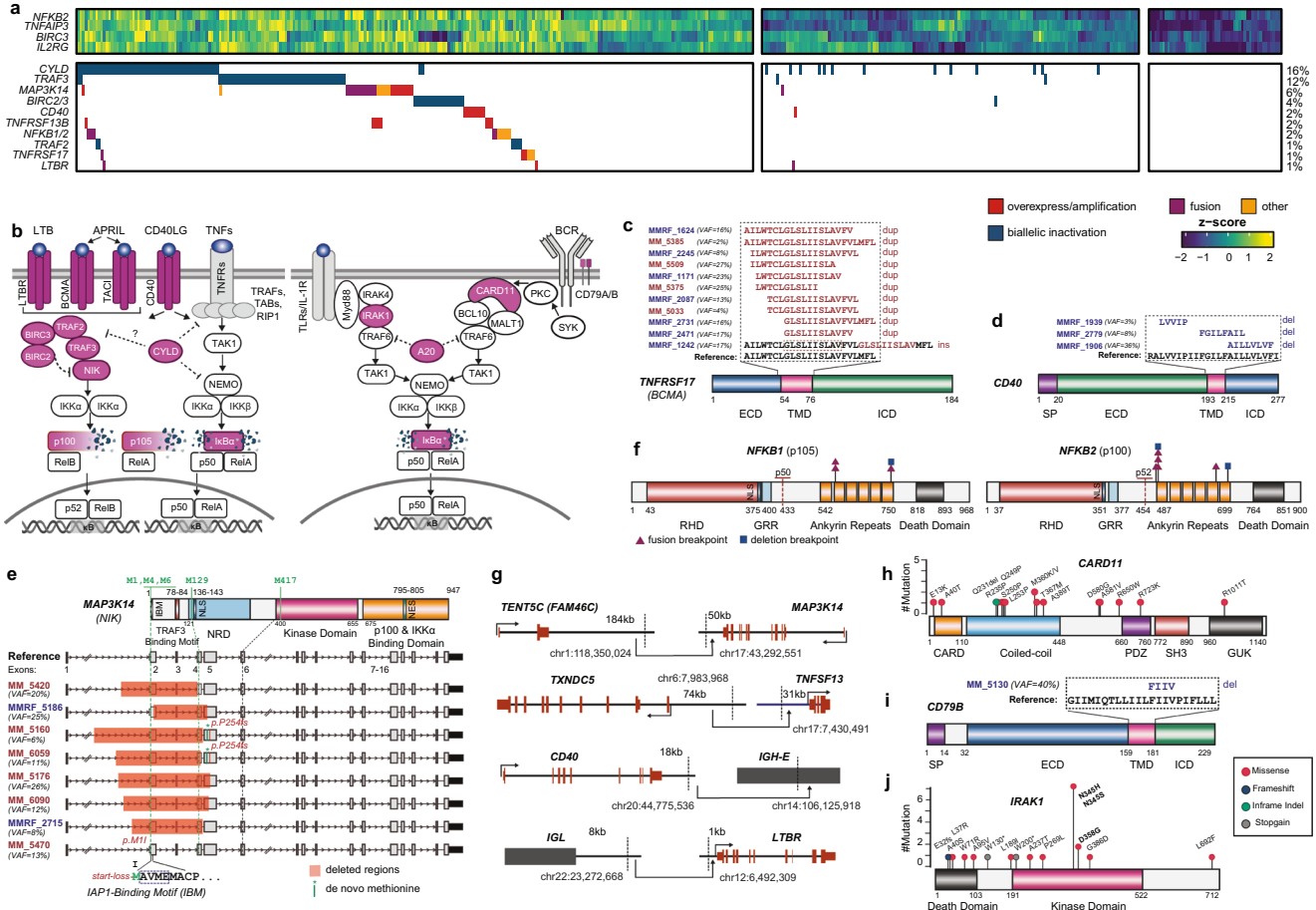

**Fig. 2 Diverse alterations of the NF-κB pathway in relapsed refractory multiple myeloma. a** Integrative heatmap for alterations in the NF-κB pathway for cases with tumor purity greater than 30% (n = 450). The transcriptomic signature for NF-κB activation was experimentally derived[16]. "Biallelic inactivation" includes homozygous deletion, hemizygous deletion coupled with mutation, or hemizygous deletion or uniparental disomy (UPD) coupled with downregulation of expression (1.5-fold below the cohort median). "Other" includes in-frame indels and internal or partial deletions. Frequency of respective alterations is provided to the right. **b** Summary of the alterations observed in the NF-κB pathway. Alterations in affected genes (highlighted in violet) were detected in all four branches of the NF-κB pathway, including TNF receptor family, non-canonical, Toll-like receptor (TLR), and B cell receptor (BCR) signaling. **c** In-frame tandem duplications or insertions in the transmembrane domain (TMD) of TNFRSF17. **d** In-frame deletions in the TMD of CD40. **e** N-terminal deletions in MAP3K14 (NIK) truncating the TRAF3 binding site in RRMM. Variant allelic fractions are indicated (VAF). **f** Schematics of gene fusions and deletions of the C-terminus of NFKB1 (left) and NFKB2 (right). **g** Translocations that lead to outlier expression of NF-κB genes, including a kinase (MAP3K14), a cytokine (TNFSF13), and cell surface receptors (CD40 and LTBR). Each translocation juxtaposed the gene of interest to a locus with a strong enhancer (IgH, IgL, FAM46C, and TXNDC5). Breakpoints are shown as dashed vertical lines. **h** Lollipop plot for CARD11 mutations in RRMM cohort. **i** In-frame deletion in the TMD of CD79B. **j** Lollipop plot for IRAK1 mutations aggregated from RRMM and newly diagnosed MM (NDMM) cohorts.

While inspecting *CD40* across the MM cohorts, we observed cases with outlier gene expression levels coincident with robust NF-κB transcriptomic signatures. This finding prompted us to perform a systematic screening for patients with rare outlier expression (Supplementary Fig. 6b) and to search for genomic rearrangements to explain these events. Our analyses revealed that cell surface receptors (*CD40*, *LTBR*) and the kinase *MAP3K14* exhibited outlier expression levels in select cases (Supplementary Fig. 6c). Overexpression of the receptors could potentially lead to oligomerization, activating downstream signaling[23]. Whole-genome sequencing of representative cases (Fig. 2g) revealed that the outlier expressed genes were translocated in proximity to loci harboring strong enhancers. Interestingly, in one patient, we observed outlier expression of the NF-κB activating ligand *TNFSF13 (APRIL)* which is associated with a translocation to the 3′ region of *TXNDC5*, a gene highly expressed in plasma cells. Although APRIL is typically secreted by monocyte-derived dendritic cells and not plasma cells[24], the tumor cells in this case ectopically express APRIL and become

more independent of the bone marrow environment in an autocrine manner. This exemplary case serves as evidence of positive selection for tumor cells that can remodel the microenvironment to facilitate their survival.

While plasma cells represent the end-stage of differentiation in B lymphocytes, a subset of MM patients shared common features with earlier stage B cell malignancies and had TLR and BCR-mediated NF-κB signaling activated by mutation (Fig. 2a, b)[25–27]. These alterations included truncating mutations in *A20* (*TNFAIP3*) and missense mutations in the coiled-coil domain of *CARD11* (Fig. 2h). An in-frame deletion in the transmembrane domain of *CD79B* was observed in one RRMM patient (Fig. 2i) which would potentially activate NF-κB in the same manner as the aforementioned in-frame indels in *TNFRSF17* and *CD40*. Interleukin-1 receptor-associated kinase 1 (*IRAK1*) exhibited loss-of-function mutations, as well as recurrent missense mutations at highly conserved residues in the kinase domain, such as Asn 345 in the catalytic loop and the magnesium ion-binding residue Asp 358 in the Asp-Phe-Gly (DFG) motif (Fig. 2j and Supplementary

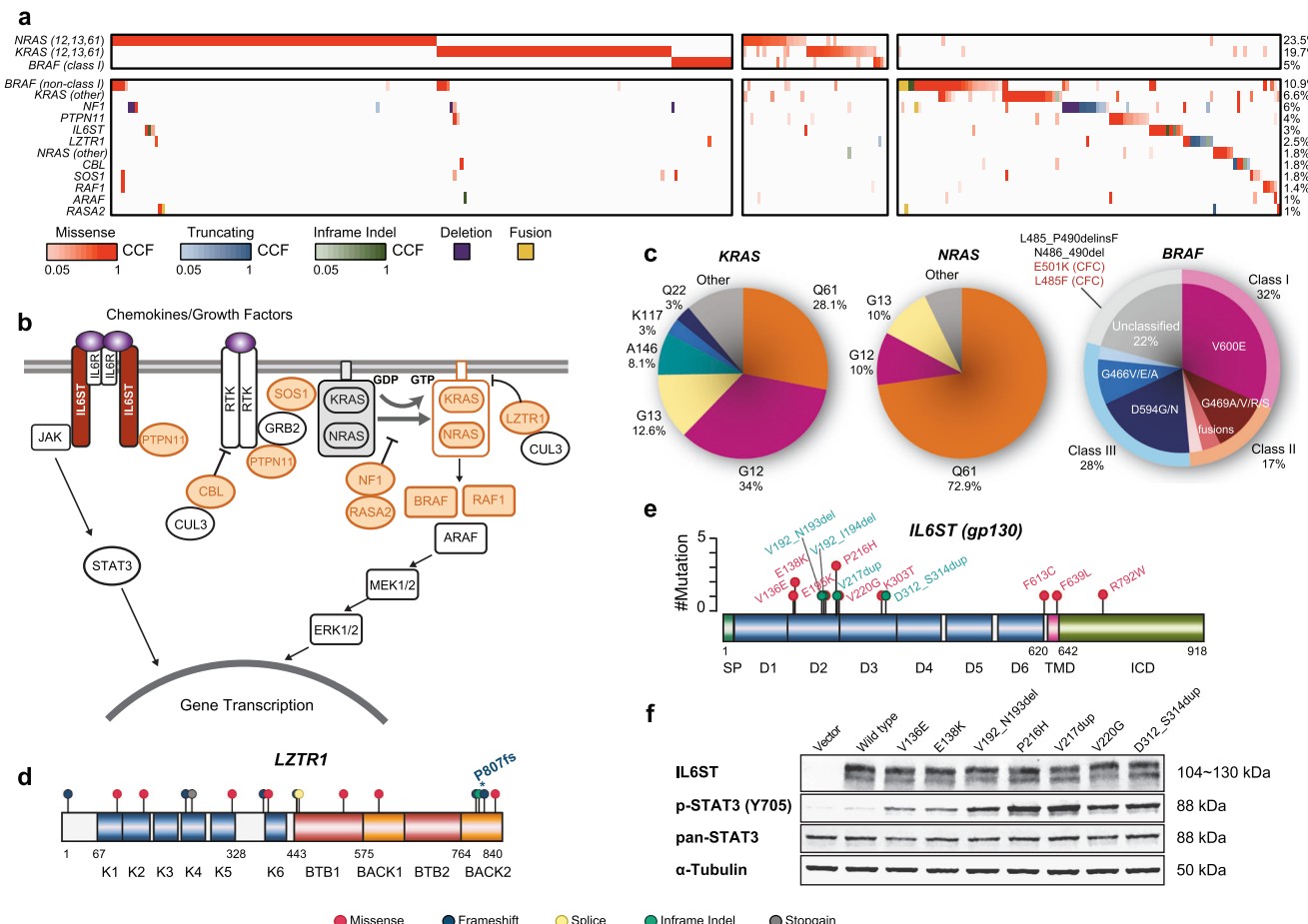

**Fig. 3 Alterations of the RAS-MAPK and JAK-STAT3 pathway in relapsed refractory multiple myeloma. a** Heatmap of RAS-RAF pathway alterations in cases with at least one mutation with CCF (cancer cell fraction) ≥0.05 (n = 354) (Methods). Clonal mutations of *RAS* and *BRAF* (*RAS* Q61, G12, G13, and *BRAF* V600) showed a strict pattern of mutual exclusivity (top left panel). Mutations which appeared to co-occur were subclonal and likely belonged to different clones (top middle panel). There was also a distinct group of cases with mutations associated with the Rasopathies (bottom right panel). *IL6ST* was also included given the reported association with SHP2 (PTPN11) **b** Overview of the RAS-RAF signaling pathway and summary of alterations observed in RRMM. Rasopathy-associated genes that had mutations in our RRMM cohort are highlighted in light orange. **c** Pie charts show the distribution of mutations in *NRAS*, *KRAS*, and *BRAF* across our RRMM cohort. **d** Lollipop plot of *LZTR1* which was enriched for loss-of-function mutations. **e** Lollipop plot of mutations in *IL6ST*. **f** JAK-STAT3 pathway activation in HEK-293FT cells overexpressing *IL6ST* mutants. Western blot analyses of protein levels of IL6ST, phosphorylated STAT3(Y705); total STAT3, and a-tubulin (loading control) are shown. The experiment was repeated twice independently with similar results. Source data are provided as a Source Data file.

Fig. 6d). Substitutions at these key residues would render the protein kinase-dead, and interestingly, have not been reported in any other hematological tumor. They also provide genetic evidence of IRAK1 functioning as a context-specific negative regulator of survival or proliferation in addition to its role in the formation of the Myddosome[28].

In summation, previous studies profiling NDMM have implicated alternative NF-κB alterations at a prevalence of 20%[17]. Our comprehensive integrative approach included all characterized NF-κB pathways and encompassed diverse mechanisms, indicating that over 60% of MM harbor alterations in this key cell survival pathway.

**Heterogeneity of alterations in the RAS-MAPK pathway including Rasopathy-related genes**. Similar to NDMM, alterations in the RAS-MAPK pathway play a major role in RRMM (Fig. 3a, b). *KRAS* was the most frequently mutated gene in RRMM with a frequency of 26%, followed by *NRAS* at 25% and *BRAF* at 15% (Fig. 1a). The distribution of mutated *KRAS* codons (Fig. 3c, left) was distinct from that observed in a pan-cancer assessment[29]. Although codon G12 was still the most common

(34%) in MM, there was strong enrichment for Q61 (28.1%) and G13 (12.6%)[29]. Across all cancers, non-G12/G13/Q61 codons constituted around 2% of total *KRAS* mutations. By contrast, the incidence for recurrent atypical codons was higher in RRMM: 8.1% for A146, 3% for K117, and 3% for Q22 (Fig. 3c, left). *NRAS* codon distribution more closely resembled the pan-cancer assessment, with Q61 as the dominant codon observed in 73% of cases, and non-G12/G13/Q61 codons in 7% (Fig. 3c, middle). Across cancers, *NRAS* G13 was twice as frequent as G12[29], while *NRAS* G12 and G13 were both observed at 10% in RRMM.

*BRAF* mutations in MM can be stratified into three classes based on their kinase activities[30,31]. All three classes of *BRAF* mutations were well-represented in our MM cohorts (Fig. 3c, right). Class 1 mutations (V600E monomer) were the most common at 32%, followed by Class 3 mutations (kinase-impaired or dead) at 28%, and Class 2 alterations (constitutive dimers, including N-terminal BRAF fusions) at 17%. Among the *BRAF* alterations whose kinase activities have not yet been classified include the in-frame deletion involving codons 485 to 490 (Fig. 3c, right). These rare in-frame indels have been demonstrated to form BRAF homodimers and are potentially

druggable[32,33]. Clonality analysis by calculating cancer cell fraction (CCF) (Methods) revealed that clonal *RAS* G12, G13, Q61, and *BRAF* V600E were strictly mutually exclusive with each other (Fig. 3a, top left cluster). It was possible to observe *RAS* G12, G13, Q61, and *BRAF* V600E in the same patient, although as subclonal events and most likely belonging to different clones (Fig. 3a, top middle cluster). The co-occurrence of many different subclonal *RAS* and *BRAF* alterations in the same patients demonstrates the high level of intraclonal heterogeneity in RRMM and emphasizes the selection pressure to activate RAS signaling in RRMM.

Interestingly, in addition to patients with *KRAS*, *NRAS*, and *BRAF* mutations, we found that MM exhibited a long tail of alterations in rare, congenital RAS-pathway-related diseases, known as the "Rasopathies" (Fig. 3a, bottom right panel, Fig. 3b). This "tail" recapitulated the spectrum of germline mutations found in Noonan syndrome (NS), cardiofaciocutaneous syndrome (CFC), LEOPARD syndrome (LS), and neurofibromatosis type 1 (NF1)[34–36]. These genetic alterations include missense mutations in the SH2 domains of *PTPN11*, missense mutations in *SOS1*, truncating mutations and focal hemizygous deletions of *NF1*, truncating mutations in negative regulators of RAS signaling such as *CBL*, *LZTR1*, and *RASA2*, and hotspot mutations in *RAF1* (S257 and S259)[34–36]. *BRAF* E501K and L485F (Fig. 3c) have also been observed as germline mutations in NS and CFC patients[37,38]. Somatic mutations in *LZTR1*, while relatively rare (2.5% with CCF ≥ 0.05), were enriched in the RRMM cohort (*P* Value < 0.01, one-sided Fisher's exact test) (Fig. 3d). LZTR1 was recently characterized as a substrate adapter for the CUL3 E3 ligase complex and can mediate the detachment of RAS from the cell membrane[39]. Germline missense and truncating mutations of *LZTR1* have been associated with NS, schwannomatosis, and pediatric brain tumors[39]. Interestingly, one of our patients had a germline frameshift at P807 (Fig. 3d), which later co-occurred with a somatic hemizygous deletion of 22q. This combination led to biallelic inactivation of *LZTR1* and likely acted as a strong RAS-activating event.

It has been reported that interleukin 6 cytokine family signal transducer (*IL6ST* or gp130) could activate the RAS-MAPK pathway through its association with PTPN11, as well in the JAK/STAT pathway through JAK[40,41]. The pattern of mutations in *IL6ST* in our RRMM cohort was strikingly similar to those described in inflammatory hepatocellular carcinoma (IHCA)[42,43]. In-frame indels and recurrent point substitutions affected the D2 domain of *IL6ST* (Fig. 3e), which could facilitate its dimerization even in the absence of IL-6[42,43]. While IHCA-associated *IL6ST* variants almost always cluster from codon 168 to 216, we observed mutations that appeared earlier in the D2 domain (V136E, E138K) and far later in the D3 domain (K303T, D312_S314dup) (Fig. 3e). These rare mutants could meditate STAT3 activation as robustly as the more common ones (Fig. 3f). In addition, *IL6ST* variants were significantly enriched in RRMM compared to NDMM (*P* < 0.001, one-sided Fisher's exact test), which reflects the progressive independence of the myeloma cells from bone marrow cytokines in some advanced patients.

**Alterations in the MYC pathway.** Dysregulation of MYC pathways is common in MM[44]. In our analyses, focal amplification and rearrangements involving the *MYC/PVT1* locus producing chimeric transcripts were detected in at least 10% of samples in the cohort (Fig. 1b and Supplementary Fig. 7a). Recurrent mutations in *MYC* were detected in the combined NDMM and RRMM cohorts, including S161L (*n* = 2 RRMM, *n* = 3 NDMM) and an uncharacterized in-frame deletion at Val 280 (*n* = 3 RRMM, *n* = 6 NDMM) (Supplementary Fig. 7b). Val 280 is located within the PEST sequence, a domain required for efficient proteolysis[45], and this alteration may disrupt post-translational modifications and enhance MYC stability[45]. *MYCL* and *MYCN* were also affected by structural rearrangements that led to over-expression (Supplementary Fig. 7c, d).

**Alterations related to disease progression and drug resistance.** We next systematically compared the mutational landscape of RRMM with NDMM from the CoMMpass cohort. We observed a significantly higher incidence of mutations and copy-number loss in tumor suppressors such as *TP53*, *RB1*, *CDKNA2/B*, *BIRC2/3*, and *CDKN2C* (Fig. 4a, b and Supplementary Fig. 8a–c), likely due to MM progression. Furthermore, in RRMM relative to NDMM, we uncovered a diverse range of mutations in genes that confer resistance to three classes of MM therapies - immunomodulatory imide drugs (iMiDs), synthetic glucocorticoids, and monoclonal antibodies (Fig. 4c–i). Immunomodulatory imide drugs (such as thalidomide, lenalidomide, and pomalidomide) function by binding to a tri-tryptophan pocket in CRBN, the substrate receptor of the CUL4–ROC1–DDB1–CRBN (CRL4CRBN) E3 ubiquitin ligase[46,47]. The binding of iMiDs induces a conformational change and shifts the target of degradation from endogenous substrates to IKZF1 and IKZF3, two essential transcription factors in MM[46,47]. Various types of alterations frequently targeted two genes of the CRL4CRBN complex in RRMM, *CRBN*, and *CUL4B*, at a cohort frequency of 11 and 5%, respectively (Fig. 4c, e, f). Missense substitutions were observed throughout the gene body of *CRBN* (Fig. 4e), including those at or near the binding pocket (E377K, H353N, G354E) and CRBN-DDB1 interface (T238N, D249Y, E187K) (Fig. 4e and Supplementary Fig. 8d, e). *CRBN* was also frequently affected by truncating mutations (frameshift, stop-gain, and splice site) and hemizygous deletion involving cytoband 3p26 (Fig. 4b and Supplementary Fig. 8c, right). *CUL4B* harbored truncating mutations and missense mutations clustered in the cullin domain, which could significantly disrupt the scaffold's structure (Fig. 4f). Acquired alterations in *CRBN* and *CUL4B* due to iMiD therapies not only have relevance to MM treatment but also presage analogous mutations that are likely to be acquired based on various proteolysis targeting chimera (PROTAC) therapies under development[48].

At high doses, glucocorticoids such as dexamethasone and prednisone affect MM cells potentially via an anti-NF-κB transcriptional program mediated by the glucocorticoid receptor (*NR3C1*)[49]. In addition to truncating mutations, missense mutations were observed in the ligand-binding domain and N-terminal domain of *NR3C1* in the RRMM cohort (Fig. 4c, g). One case had an in-frame intragenic deletion that removed the DNA-binding domain of *NR3C1* (Supplementary Fig. 9a). This deletion may disrupt NR3C1's transcriptional activity while retaining protein interactions, such as binding to co-activators. Interestingly, retinoic acid receptor alpha (*RARA*), another nuclear receptor, harbored mutations enriched in the RRMM cohort compared to the NDMM cohort (Fig. 4a). *RARA* mutations clustered in the ligand-binding domain (Fig. 4h), a pattern similar to those found in fibroepithelial breast tumors[50]. It has been demonstrated that these mutations may not disrupt RARA's ligand-binding ability but enhance its interaction with other co-repressors[50].

Monoclonal antibodies targeting CD38 may induce cell death via antibody-dependent cellular phagocytosis (ADCP) or antibody-dependent cellular cytotoxicity (ADCC)[51]. In RRMM relative to NDMM, we screened for alterations in *CD38* and identified several loss-of-function events, such as homozygous deletions, truncating mutations, and fusions in which the

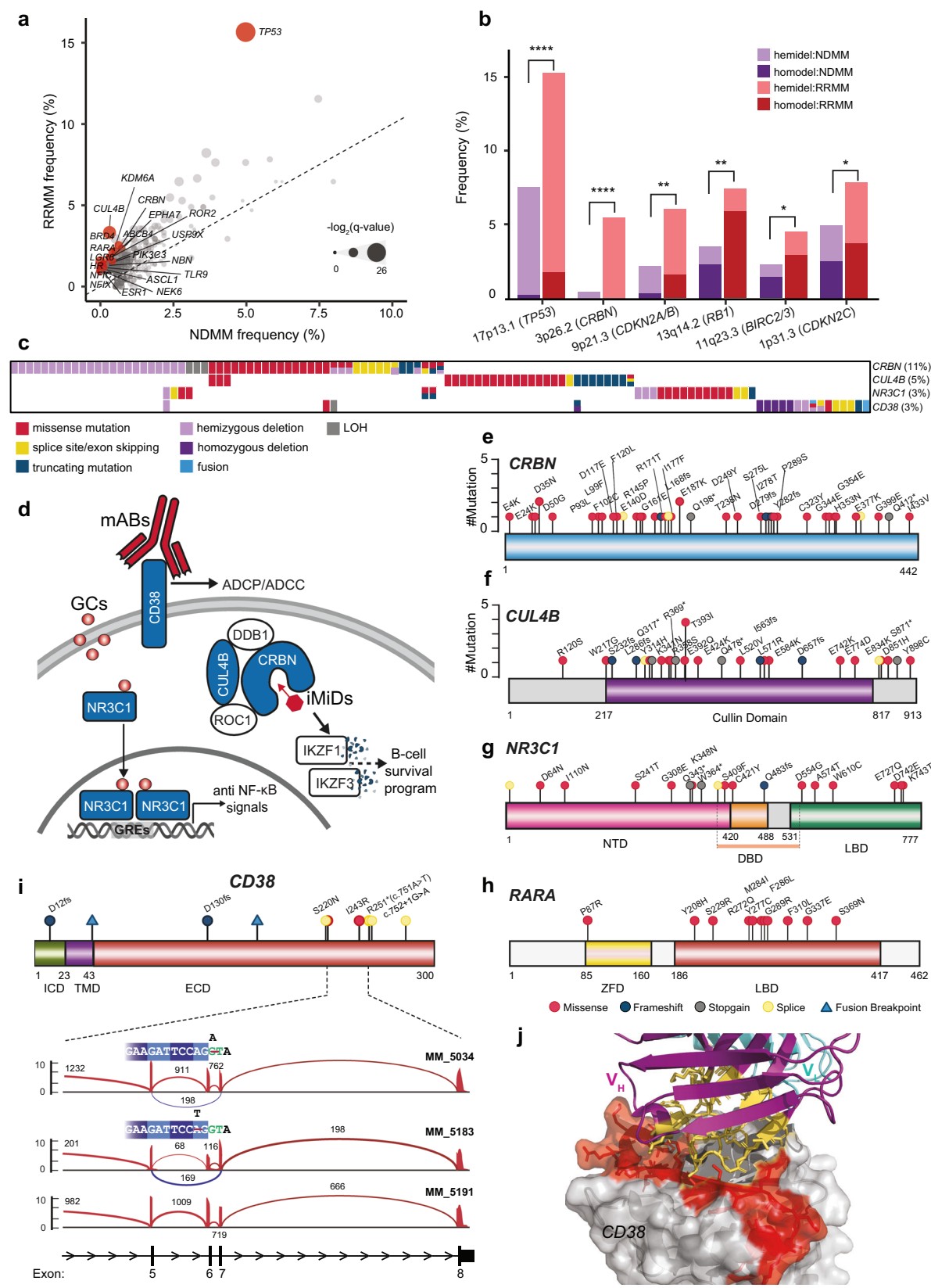

extracellular domain was lost (Fig. 4c, i, top, Supplementary Fig. 9b, c). Systematic analysis of de novo splice junctions identified two cases with exon 6 skipping (Fig. 4i, bottom). One case exhibited a splice donor mutation, while another had a point substitution of two nucleotides upstream of the splice site. This point mutation could function as a stop-gain (R251*) but instead

unexpectedly induced in-frame exon skipping and removed most of the epitopes on CD38 that interact with daratumumab[52]. These events may facilitate the evasion of MM cells from binding by daratumumab, while still retaining a major portion of the extracellular domain (Fig. 4i, j). Overall, CD38 mutations were associated with lower CD38 expression than the rest of the cohort

**Fig. 4 Alterations enriched in relapsed refractory multiple myeloma associated with drug resistance or disease progression. a** Scatter plot of mutation frequency in RRMM vs. NDMM. The size of the circles correlates with log2 of $q$ value two-sided Fisher exact test. Genes with top coefficients from the regression model (Methods) are highlighted in red. **b** Comparison of focal deletions (less than 20MB) in NDMM vs. RRMM**. \*, \*\*, \*\*\*** indicate $P < 0.05$, 0.01, 0.001, respectively for the two-sided Fisher exact test. $P$ values obtained: $7.1 \times 10^{-6}$ (*TP53*), $1.4 \times 10^{-9}$ (*CRBN*), $1.3 \times 10^{-4}$ (*CDKN2A/B*), $2.2 \times 10^{-3}$ (*RB1*), 0.025 (*BIRC2/3*), and 0.035 (*CDKN2C*). **c** Heatmap of drug resistance-related genes identified in RRMM. The types of alterations are indicated in the legend. There were cases with alterations observed in more than two genes or different types of alterations per gene, reflecting the complex history of tumor evolution through several treatments. **d** Overview of therapies in MM with observed resistance mechanisms. Genes highlighted in blue are resistance mutations found in RRMM. ADCP antibody-dependent cellular phagocytosis, ADCC antibody-dependent cellular cytotoxicity, mABs monoclonal antibodies, GCs glucocorticoids, iMiDs immunomodulatory imide drugs. **e–h** Lollipop plots for *CRBN, CUL4B, NR3C1*, and *RARA*. **i** Top panel, lollipop plot for *CD38*. Frameshift mutations would completely abolish CD38, and fusions that truncate the extracellular domain would disrupt binding events. Bottom panel, Sashimi plot for two cases with exon 6 skipping. MM_5034 had a splice donor mutation. MM_5183 had a missense mutation that was two nucleotides upstream of the splice site, which would naturally introduce a new stop codon (R251\*). Unexpectedly, this mutation functioned as a splice donor mutation instead, which also induced the skipping of exon 6. MM_5191 is included as a control. **j** The skipping of *CD38* exon 6 in MM_5034 and MM_5183 was in-frame and would delete 29 amino acids (highlighted in red), including the epitope of daratumumab (PDB structure 7DUO)[52].

(one-sided Wilcoxon rank-sum test $P < 2 \times 10^{-3}$) (Supplementary Fig. 9d).

While many mutations causing resistance were detected at low variant allele frequency (VAF), some appeared "clonal" due to a bottleneck effect, where a single cell that acquired resistance mutations survives treatments and becomes the "founder" (Supplementary Fig. 9e, f). Several RRMM patients also acquired resistance-related mutations in multiple genes (Fig. 4c and Supplementary Fig. 9f), reflecting complex histories of tumor evolution under varied treatments. The intra- and inter-clonal heterogeneity of RRMM can also be inferred by the presence of CTCs in peripheral blood. For example, one sample harbored a stop-gain mutation of *CRBN* detected only in the peripheral blood but *not* at the biopsy site. However, copy-number analysis of the bone marrow aspirate specimen revealed a subclonal hemizygous deletion of *CRBN* (Supplementary Fig. 9g). These spatially separated alterations were the result of convergent evolution resulting in resistance to iMiD treatment.

## Discussion

RRMM is difficult to manage malignancy with an aggressive disease course[3,4]. Our study represents comprehensive integrative analyses of RRMM genetics and, further, systematically compares the genetic landscape of RRMM with that of NDMM. By covering somatic mutations, indels, copy-number alterations, gene fusions, gene expression, and outlier expression, we demonstrate that a majority of MM patients harbor alterations in the NF-κB and RAS-MAPK pathways; this is a level of prevalence that has not been previously reported and includes many alterations with diverse mechanisms of action.

It was surprising to observe that at least 45% of MM harbors alterations in the NF-κB pathway. Several activating mechanisms were identified, including: (1) in-frame insertions and deletions in the transmembrane domain of *TNFRSF17* and *CD40*, (2) N-terminal intragenic deletions of *MAP3K14*, (3) in-frame C-terminal fusions and deletions in *NFKB2* (p100) and *NFKB1* (p105), (4) genomic rearrangements or translocations of *CD40, LTBR*, and *TNFSF13*, and (5) diverse alterations in *A20, CARD11, CD79B*, and *IRAK1*. Notably, mid-size to larger indels in *CD40, TNFRSF17*, and *MAP3K14* already occurred in the published CoMMpass dataset but have not been reported by any studies. These results highlighted the importance of thorough bioinformatics analyses in variant calling. It is fascinating to note that recurrent mutations in *IRAK1* were kinase-dead mutants in MM. Our finding agrees with a previous study that showed IRAK1 is an important upstream adapter for NF-κB signaling via TLRs, but its kinase domain is dispensable for signaling activity[53]. It is also possible, as is the case for another receptor-associated kinase of

the NF-κB pathway, RIPK1, that the kinase domain of IRAK1 plays a role in inducing apoptosis[54].

Alterations in the RAS-MAPK pathway in MM are even more prevalent than alterations in the NF-κB pathway. In addition to the well-characterized *NRAS, KRAS*, and *BRAF* genes, our study revealed that the germline Rasopathy genes represent a long tail of somatic alterations linking MM to these rare, congenital RAS-pathway-related diseases. If the designation of Rasopathy is extended to include mosaic conditions, such as keratinocytic epidermal nevus syndrome[55], alterations in *FGFR3* can also be integrated into this long tail, making alterations in the RAS-MAPK pathway even more prevalent in RRMM. It is generally observed that the spectrum of mutations in the Rasopathies and in cancer minimally overlap, as exemplified by *PTPN11* and *BRAF*[56,57]. One possible explanation is that cancer-associated RAS-MAPK mutations would be lethal for embryonic development, while the Rasopathy-associated RAS-MAPK mutations are too mild to evade apoptosis in malignant transformation. Interestingly, RAS-MAPK aberrations in RRMM are a conglomeration of both, making RRMM an ideal model to study strong and weak RAS-activating events. Future studies should investigate the correlation between strong and weak RAS activators with the clinical history and outcome of RRMM patients

We observed that *IL6ST* aberrations are enriched in RRMM, suggesting that this gene is associated with MM progression. *IL6ST* engages in both RAS-MAPK signaling via PTPN11 and JAK/STAT3 signaling via JAK[40,41]. As in inflammatory hepato-cellular carcinoma, point mutations and in-frame indels of *IL6ST* occurred within or nearby the dimerization interface[42,43] and could activate STAT3. Follow-up studies could investigate the use of approved JAK inhibitors, like ruxolitinib and tofacitinib, as potential therapeutic strategies for a subset of relapsed refractory multiple myeloma patients.

A unique opportunity afforded by this study was the ability to systematically compare the MMRF's multi-institutional cohorts of NDMM patients with RRMM patients sequenced at our center. In addition to *IL6ST* mentioned above, mutations and copy-number loss in tumor suppressors such as *TP53, RB1, CDKNA2/B, BIRC2/3*, and *CDKN2C* were enriched in RRMM, suggesting that these are also likely events associated with disease progression. Our analyses also identified resistance alterations that develop in the context of commonly used MM therapies, including iMiDs (*CRBN* and *CUL4B*), synthetic glucocorticoids (*NR3C1*), and monoclonal antibodies (*CD38*), which expands upon previous studies[58,59]. The most common resistance alterations were associated with iMiDs, and these alterations have significant implications for clinical research and drug development beyond MM. For example, given that CRBN-based PRO-TACs are emerging as a promising approach for targeted

therapy[60], developments in this drug class must anticipate possible resistance via acquired alterations in *CRBN* and *CUL4B*. Indeed, in vitro CRISPR-Cas9-mediated loss-of-function screening in prostate cancer cells treated with a CRBN-based BRD4 degrader also predicted CRBN as the most likely candidate for drug resistance[48]. Furthermore, as mutations in *CRBN*, *CUL4B*, and *NR3C1* were detected in peripheral blood in a subset of RRMM patients, these genes should be included in CTC screening panels employing next-generation sequencing. Interestingly, despite our unbiased effort to search for the enrichment of mutations and copy-number alterations in RRMM compared to NDMM, a genetic basis for resistance against proteasome inhibitor drug class remains elusive. We suggest that genome-wide gain-of-function or loss-of-function CRISPR screening could help to narrow the candidate gene list.

Another clinically relevant aspect of our study concerns the monoclonal antibodies targeting CD38. Although potent as MM therapies, they were subject to resistance mutations and copy losses, according to our analyses. Two of our patients harbored distinct mutations that converged into the in-frame exon skipping affected codons 221 to 250, presumably only disrupting the epitope of daratumumab while retaining a major portion of the extracellular domain. In theory, such patients could still benefit from isatuximab, another monoclonal therapy targeting CD38. As ref. [52] pointed out, that the epitope of isatuximab is composed of residues from codons 34 to 189, thus completely unaffected by this exon skipping the event. Future structural and clinical studies should explore this direction to widen the therapeutic options for patients who relapse on daratumumab. Together, this study defines the genetic landscape and pathways of progression to RRMM while also identifying targeted therapy resistance mechanisms likely to impact the clinical management of RRMM.

## Methods

**Patient samples**. All samples were acquired after patients provided written informed consent in compliance with the Multiple Myeloma Research Foundation Institutional Review Board (IRB) (Protocol# MMRF-002; IRB Tracking Number 20151186) and the University of Michigan IRB. Germline assessment was carried out using blood specimens. CD138+ cells from fresh bone marrow mononuclear cells were isolated by immuno-magnetic selection. In brief, mononuclear cells were separated from bone marrow by density centrifugation by Ficoll-Paque (GE Healthcare). Mononuclear cells were then incubated with CD138 antibody-conjugated microbeads and loaded onto a MACS column according to the manufacturer's instructions (Miltenyi Biotec). After washing with 10 volumes of PBS buffer, the column was removed from the magnetic stand, and CD138+ cells were eluted from the column. Smears of CD138-selected cells were prepared by cytospin centrifuge and stained by HEMA-DIFF Fixative/Xanthene/Thiazine reagents (StatLab Medical Products) for pathology review. Samples passing tumor content assessment were processed for sequencing analysis.

**Integrative clinical sequencing**. Integrative clinical sequencing was performed using standard protocols in our Clinical Laboratory Improvement Amendments (CLIA)-certified sequencing laboratory. CD138 + tumor cells and matched-normal blood mononuclear cells were resuspended in RLT lysis buffer (Qiagen) and disrupted by 5 mm beads on a Tissuelyser II (Qiagen). Genomic DNA and total RNA were purified from the same sample using the AllPrep DNA/RNA/miRNA kit (Qiagen). Matched-normal genomic DNA from blood, buccal swab, or saliva was isolated using a DNeasy Blood & Tissue Kit (Qiagen). RNA integrity was measured on an Agilent 2100 Bioanalyzer using RNA Nano reagents (Agilent Technologies). RNA-sequencing was performed by the exome-capture transcriptome platform developed in our lab and as described previously[61]. In brief, capture transcriptome libraries were prepared using 1–2 µg of total RNA. Following the steps of cDNA synthesis, end-repair, A-base addition, and ligation of adapters, pre-capture libraries were size-selected by the PippenHT system (Sage Science). Recovered fragments were enriched by PCR using Phusion DNA polymerase (New England Biolabs) and index primers and purified by AMPure XP beads (Beckman Coulter). Coding exons were then captured by Agilent SureSelect Human All Exon v.4 probes following the manufacturer's protocol. Final sequencing libraries were analyzed by Agilent 2100 Bioanalyzer for product size and concentration. Libraries were sequenced by the Illumina HiSeq 2500 (2 × 126-nucleotide read length), with

a sequencing coverage of 40–60 million paired reads. Reads that passed the chastity filter of Illumina BaseCall software were used for subsequent analysis.

Exome libraries of matched pairs of tumor/normal DNA were prepared as previously described[9]. In brief, 1 µg of genomic DNA was sheared using a Covaris S2 (Covaris) to a peak target size of 250 bp. Fragmented DNA was purified using AMPure beads, followed by end-repair, A-base addition, and ligation of adapters using the Kapa HyperPrep kit and protocols (Roche/Kapa Biosystems). DNA molecules were size-selected by the PippenHT system (Sage Science). Fragments between 300–350 bp were recovered, amplified by KAPA HiFi HotStart Mix and index primers, and purified by AMPure beads. One microgram of the pre-capture library was hybridized to an in-house developed Oncoseq targeted gene panel containing 1711 genes with suggestive links to cancer (probes synthesized by Roche). The targeted exon fragments were captured and enriched following the manufacturer's protocol (Roche). Final sequencing libraries were analyzed by Agilent 2100 Bioanalyzer and sequenced by Illumina HiSeq 2500 (Illumina; 2 × 126-nucleotide read length).

FastQC[62] (version 0.11.8) was used to assess read quality per lane. FASTQ conversion was performed with bcl2fastq-1.8.4 in the Illumina CASAVA 1.8 pipeline. Picard (version 2.20.3) was used to monitor other sequencing metrics such as duplication rate, GC biases, and targeted coverage.

**Alignment, mutation calling, and filtering**. The FASTQ files were aligned to the reference genome build hg19/GRCh37 using Novoalign[63] Multithreaded (version 3.02.08) (Novocraft) and converted into BAM files using Samtools[64] (version 0.1.19). BAM files were sorted, indexed, and marked duplicates with Novosort (version 1.03.02). Single nucleotide variants (SNVs) and small indels were called by freebayes[65] (version 1.0.1). Larger indels and exon-level structural arrangements were called with pindel[66] (version 0.2.5b9). freebayes and pindel calls were then compiled and annotated to RefSeq and COSMIC[67] v90, dbSNP[68] v146, ExAC[69] v0.3, and 1000 Genomes phase 3[70] databases using snpEff[71] (version 4.3t) and snpSift[72] (version 4.3t).

We employed a two-step filtering strategy to detect somatic mutations when there was contamination of CTCs in peripheral blood controls. First, mutation calling was performed on normal libraries to estimate the level of contamination. This information would be used to adjust the somatic filtering threshold accordingly. To be more specific:

- In the normal-only variant filtering step, a normal-contaminated mutation was identified as the call that: (1) was supported by at least five reads, (2) had the population minor allele frequency (MAF) <0.05%, (3) passed the filtering against an in-house database of recurrent sequencing artifacts (e.g., pool-filtering) constructed from 2000 genomic sequencing libraries. Known hotspot mutations (such as those from *TP53*, *NRAS*, *KRAS*, *BRAF*, *PIK3CA*, etc.) were inferred from COSMIC v90 and 2000 genomic libraries of the same sequencing platform at our center, then "white-listed" out from the pool-normal. This is because variants at these locations are more likely to be real mutations than artifacts. To enhance the specificity, variants with ultra-low Variation Allelic Fraction VAF (<5%) were (4) further filtered for 8'oxoG artifacts[73], (5) required to have evidence supported by reads in both forward- and reverse- strands. Strand-specific reads at a given position were counted by bam-readcount[74] (version 0.8.0) for SNV and small indel up to 5 bp. The maximum VAF of all contaminated mutations in each normal library was then used as a threshold to filter somatic mutations in the next step. This threshold could be more than 30% for some cases in our cohort (Supplementary Fig. 2d). This approach allowed us to discover CTC-associated mutations that may not even be present at the biopsy site (Supplementary Fig. 9g).
- In the tumor-normal variant filtering step, in addition to satisfying criteria (1–5) above, a somatic mutation should tolerate a matched-normal VAF up to the contamination threshold (if there is no normal contamination, an arbitrary threshold of 2% was applied, Supplementary Fig. 2d, horizontal dashed line). To further distinguish recurrent indel artifacts in homopolymer regions from true somatic variants, especially in tumors with microsatellite instability phenotype, a logistic regression model using PCR duplication rate as a covariate to model variant and total read counts were applied to improve the read cut-off for indels, as detailed previously[75].

Variant calling for SNVs and small indels from RNA-seq (tumor-only) was performed using sentieon[76] (version 202010.02) (Sentieon, Inc). Calls were annotated to RefSeq and COSMIC67 v90, dbSNP68 v146, ExAC69 v0.3, and 1000 Genomes phase 3. We adopted a similar variant filtering strategy to the normal-only variant filtering step detailed above, and further filtered calls against RADAR[77], a database for A-to-I RNA editing. Potential variants in *CD38* were validated by whole-exome sequencing using Agilent Human All Exon v4 reagents.

**Copy-number analysis**. Targeted sequencing and whole-exome sequencing data were analyzed for copy-number using an in-house pipeline (cnatools), as previously described[75]. Circular binary segmentation (CBS) algorithm (as implemented in DNAcopy[78] version 1.48.0) was used to jointly segment B-allele frequencies and log2-transformed tumor/normal coverage ratios across targeted

regions. The expectation-maximization (EM) algorithm[79] was used to jointly estimate tumor purity and classify regions by copy-number status, namely gain, loss, and loss of heterozygosity. CNVkit[80] (version 0.9.6) with CBS and haar segmentation method were also run in all samples. Results from cnatools and CNVkit were compiled by a customized script to inspect discrepancies in segmentation, especially for highly heterogeneous samples with subclonal CNAs. For each result from cnatools, different ploidy models were manually reviewed by D.R.R., Y.-M.W., and J.N.V. to account for the possibility of whole-genome duplication (tetraploidy).

GISTIC2.0[81] was used to locate arm-level and focal peaks of recurrent copy-number gains and losses. Germline CNAs, IgH, IgL, IgK, TCR loci, and recurrent noisy segments were removed before running GISTIC2.0. Significant peaks were defined as those with a q value < 0.05.

**Trinucleotide mutational analysis**. The limited number of mutations detected from targeted sequencing in our study did not allow us to discover de novo trinucleotide mutational signatures. However, APOBEC-enriched patients can be assessed as in Roberts et al.[82]. For each case, within a fraction of the captured genome, the number of C-to-T or C-to-G, and G-to-A or G-to-C substitutions that occurred in and out of the APOBEC motif (**TCW** or **W**GA), as well as the total number of C or G occurred in and out of APOBEC motif, was tabulated. One-sided Fisher's exact tests were performed, and P values were corrected using the Benjamini–Hochberg method. Samples with FDR < 0.05 were marked as APOBEC-enriched.

**Gene expression and fusion analysis**. Strand-specific RNA-seq FASTQ files were aligned to reference genome build hg19/GRCh37 in chimeric alignment mode by STAR aligner[83] (version 2.7.4a). After alignment, libraries with ribosomal content ≥60% mapped reads (i.e., failed ribosomal removal) and libraries with a low number of splice junctions (<25th percentile of all in-house libraries) were excluded from the final cohort. Gene expression was quantified with featureCounts[84] (version 2.0.0), and gene fusions were called using an in-house pipeline as previously described[75]. Highly recurrent RNA chimeric transcripts (e.g., "trans-splicing") were filtered out from the reported fusions (Supplementary Data 4). Differential gene expression analysis was performed with edgeR[85] (version 3.34.0).

**CoMMpass data re-analysis**. Using CoMMpass consortium data, we reanalyzed raw (FASTQ) NGS data using our integrative pipeline so that direct comparisons could be made to RRMM samples sequenced in this study. FASTQ files were downloaded from dbGaP with the accession number phs000748 and analyzed for mutation and copy-number, as outlined above. We adjusted critical parameters and reference files as CoMMpass used a different probe set for whole-exome sequencing (Agilent Capture V5 + 5′UTR). After excluding duplicate entries, and samples without available matched peripheral blood normal or CD3 + selection normal, the remaining samples were included for mutation and copy-number analyses. A small number of CoMMpass samples were subjected to whole-exome sequencing more than once. To fairly compare mutation and CNA incidence, we only included the cases with the highest number of mutations. Finally, we excluded samples that did not pass quality control assessed from Picard's AT-dropout metrics, which resulted in noisy copy-number results.

**Nomination of cancer driver genes panel**. Integration of results from various statistical tools that predict significantly mutated genes would result in a more comprehensive list of key cancer driver genes since these tools are complementary[11]. We, therefore, ran oncodriveFML[86] (version 2.0.3), oncodriveCLUSTL[87] (version 1.1.3), MutsigCV[88], Mutsig2CV[89], and 20/20+[90] (version 1.0.1) on RRMM SNV and small indels. A relaxed threshold FDR <0.2 was applied in each tool to call significantly mutated genes. Finally, we only included genes that were predicted to be significantly mutated in at least two different statistical tools (Supplementary Fig. 2a).

**Outlier expression analysis**. To search for rare outlier expression, we used the statistical approach employed by the R package OUTRIDER[91] (version 1.7.1), which models RNA-seq read counts with a negative binomial distribution and corrects for variations in sequencing depth and co-variations across samples. Before parameter fitting, OUTRIDER requires the control of confounders, which was performed by PEER[92] implementation on 100 factors. Finally, genes with outliers were prioritized by the minimum p value (or maximum log p value) among all detected samples.

**Clonality analysis**. The cancer cell fraction (CCF) of a variant (including point mutation or small indel) i was defined as in ref. [93]. Briefly, the relationship between mutation multiplicity $m_i$ of a variant and its cancer cell fraction $CCF_i$ is considered as the following:

$$u_i = CCF_i m_i \tag{1}$$

where:

$$u_i = \frac{(1 - \text{purity}) * 2 + \text{purity} * \text{localcopynumber}_i}{\text{purity}} VAF_i \tag{2}$$

Ideally, a clonal mutation should have a CCF of 1.0 (100% of tumor cells should contain this mutation), and a subclonal mutation should have a CCF less than 1.0. Therefore, the multiplicity $m_i$ can be calculated as:

$$m_i = \begin{cases} u_i, & u_i \geq 1 \\ 1, & u_i < 1 \end{cases} \tag{3}$$

To account for the uncertainty in the estimation of tumor purity and local copy-number, a clonal mutation was defined as one with CCF ≥0.8, whereas a subclonal mutation had CCF <0.8. In one illustrative case (Supplementary Fig. 9f), SciClone[94] was used to perform 1-D clustering of VAF of mutations on diploid regions.

**Structural rearrangement analyses for whole-genome sequencing**. Whole-genome sequencing FASTQ files were aligned to the reference genome build hg19/GRCh37 with bwa-mem[95] (version 0.7.17). BAM files were sorted, indexed with Samtools, and marked duplicates with samblaster[96] (version 0.1.25). Structural rearrangements such as chromosomal translocations and deletions were called with LUMPY[97] (version 0.3.1). Split reads supporting the breakpoints were further confirmed by Blast[98].

**Comparing point mutation frequencies between cohorts**. The number of patients affected by point mutations in the RRMM and NDMM cohorts was tabulated for each gene in the Onco1700 panels. Functionally, point mutations can be synonymous, missense, stop-gain, start-loss, and splice site. Two-sided Fisher exact tests were performed, and FDR was calculated using the Benjamini–Hochberg method. Since the sequencing coverages for CoMMpass WES data were lower than the coverages for our targeted sequencing, we only included mutations with VAF ≥5% in the comparison.

We also used another regression-based strategy to account for the observations that mutation rates varied between samples and cohorts, and some patients harbored several mutations per gene. Mutation counts for genes in the Onco1700 panels were considered a set of markers used to assess their association with a cohort (RRMM vs. NDMM) and derive an optimal gene set to determine cohort memberships. All available markers were considered potential candidates for an optimal predictive signature. Since the number of candidates was high compared to the number of subjects, regularization methods based on logistic regression were used. The individual mutation rate was used as the covariate. LASSO and Elastic Nets penalty[99] were applied, and a full regularization path was computed. The choice of the optimal regularization parameter was done by maximizing the area (AUC) under the receiver-operating characteristic curve (ROC) as a criterion (Supplementary Fig. 8b). Tenfold cross-validation was used to correct for over-optimistic model-building bias. An average over cross-validation run was reported in the final ROC analysis predicting the performance of the marker signature in future observations. LASSO penalty was preferred for its ability to drop non-essential markers from the signature by explicitly assigning them zero weights. Genes with top coefficients were included in Supplementary Data 5.

**Data visualization**. Molecular graphics were generated with Pymol[100] (version 1.8.2) and some illustrations were created using BioRender.com.

**NF-κB reporter assay**. The impact of mutations in CD40 and TNFRSF17 on the NF-κB signaling pathway was assessed by NF-κB reporter assays. HEK-293FT cells (purchased from the ThermoFisher/Invitrogen) were plated in 24-well plates at a density of $10^5$ cells per well in Dulbecco's Modified Eagle medium (DMEM) containing 10% fetal bovine serum and antibiotics. The next day, cells were transiently co-transfected with CD40 or TNFRSF17 wt and mutant expression plasmids at 250 ng/well, the pGL4.32[luc2P/NF-κB-RE/Hygro] reporter plasmid at 250 ng/well (Promega), and the pRL-TK internal control plasmid at 25 ng/well (Promega) using the FuGene-HD transfection protocol (Promega). The pGL4.32[luc2P/NF-κB-RE/Hygro] plasmid contains five copies of an NF-κB response element that drives the luciferase reporter gene transcription. Cells were harvested 48 hours post-transfection, and luciferase activities were measured using the Dual-Luciferase Reporter Assay System (Promega).

**Functional analysis of IL6ST variants**. The full-length open reading frame of the wild type IL6ST was generated by PCR. Patient-derived IL6ST mutations were subsequently generated by site-directed mutagenesis (QuikChange, Agilent). IL6ST variants were cloned in the lentiviral vector pCDH510 (System Biosciences) for mammalian expression. Expression constructs were transfected into HEK-293FT cells using FuGene-HD transfection reagent (Promega). Transfected cells were cultured in DMEM medium supplemented with 10% FBS for 48 h and harvested for Western blot analysis. HEK-293FT was purchased from the ThermoFisher/Invitrogen and validated by genotyping. Antibodies and their commercial sources are as follows: anti-IL6ST/gp130 (Abcam, ab283685, 1:1,000 dilution), anti-alpha-tubulin (Abcam, ab184577, 1:5,000 dilution), anti-phospho-STAT3-Y705 (Cell

Signaling, 9131 S, 1:2,000 dilution), and anti-STAT3 (Cell Signaling, 4904 S, 1:1,000 dilution).

**Reporting Summary**. Further information on research design is available in the Nature Research Reporting Summary linked to this article.

## Data availability

All raw sequencing data (fastq files of targeted sequencing and RNA-seq) from RRMM patients enrolled in this study have been deposited in the database of Genotypes and Phenotypes (dbGaP) under accession number phs002498.v1.p1. Raw sequencing data (fastq files of WES and RNA-seq) of the CoMMpass study can be accessed from dbGaP under accession number phs000748.v7.p4. Per dbGaP policy, these datasets are available under controlled access since they contain de-identified individual-level genotype and phenotype information. Principal investigators wishing to access these data must submit their dbGaP Access Applications through the NCBI dbGaP website. Access to these datasets must be renewed annually. Additional information about the application process can be found on dbGaP website. All structures used in the analysis (7DUO[52] and 4CI2[46]) are available on PDB. The remaining data are available within the Article, Supplementary Information, or Source Data file. Source data are provided with this paper.

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

## Acknowledgements

This work was primarily supported by the Multiple Myeloma Research Foundation (ORSP 19-PAF05802). Ancillary support was provided by the National Cancer Institute (NCI, grant R35-CA231996), the Early Detection Research Network (grant U01-CA214170), and the Prostate Cancer Foundation. J.N.V. is supported by funding from the NCI Clinical Proteomic Tumor Analysis Consortium (grant U24-CA210967). A.M.C. is a Howard Hughes Medical Institute Investigator, A. Alfred Taubman Scholar, and American Cancer Society Professor. We thank Stephanie Ellison for her assistance in manuscript review and editing. We also thank Yelena Kleyman-Smith, Jin Chen, and Pankaj Vats for assistance in cloud analytics and other computational analyses. We acknowledge the efforts of the Mi-Oncoseq clinical sequencing team. The authors would also like to recognize the tremendous generosity and selflessness of the multiple myeloma patients and their families for participating in these studies.

## Author contributions

J.M., S.H., Y.N., J.Z., J.C.E., X.C., D.R.R., and Y.-M.W. coordinated clinical sequencing. Computational and bioinformatics analyses were carried out by J.N.V., D.R.R., Y.-M.W., A.C.H., W.K.B.C., J.C.E., and A.R. Functional testing of mutations was performed by S.H., J.M., D.R.R., and Y.-M.W. Hematopathological analyses were carried out by R.M., Y.-M.W., and L.W. A.T. coordinated biostatistical analyses. Y.-M.W. and D.R.R. prepared the clinical reports for this study. H.J.C., D.A., and J.Y. provided foundation support, administration, and coordination of this study. P.L.B., J.C.Y., C.E.C., M.T., and the MMRC coordinated the clinical protocol and enrollment of RRMM patients. J.N.V., D.R.R., Y.-M.W., and A.M.C. developed the figures and tables and wrote the manuscript with input from all authors. D.R.R. and A.M.C. designed and supervised the study.

## Competing interests

The authors declare no competing interests.

## Additional information

## Multiple Myeloma Research Consortium

Sikander Ailawadhi[16], Jesus G. Berdeja[17], Craig C. Hofmeister[18], Sundar Jagannath[19], Andrzej Jakubowiak[20], Amrita Krishnan[21], Shaji Kumar[22], Moshe Yair Levy[23], Sagar Lonial[18], Gregory J. Orloff[24], David Siegel[25], Suzanne Trudel[26], Saad Z. Usmani[27], Ravi Vij[28], Jeffrey L. Wolf[29] & Jeffrey A. Zonder[30]

[16]Division of Hematology/Oncology, Mayo Clinic, Jacksonville, FL, USA. [17]Sarah Cannon Research Institute, Nashville, TN, USA. [18]Department of Hematology and Medical Oncology, Winship Cancer Institute of Emory University, Atlanta, GA, USA. [19]Tisch Cancer Institute, Icahn School of Medicine at Mount Sinai, New York, NY, USA. [20]University of Chicago Medical Center, Chicago, IL, USA. [21]Judy and Bernard Briskin Center for Multiple Myeloma Research, Department of Hematology and Hematopoietic Cell Transplantation, City of Hope, Duarte, CA, USA. [22]Department of Medicine, Division of Hematology, Mayo Clinic, Rochester, MN, USA. [23]Baylor University Medical Center, Dallas, TX, USA. [24]Virginia Cancer Specialists, Fairfax, VA, USA. [25]John Theurer Cancer Center, Hackensack University Medical Center, Hackensack, NJ, USA. [26]Division of Medical Oncology and Hematology at Princess Margaret Cancer Centre, Toronto, ON, Canada. [27]Myeloma Service, Department of Medicine, Memorial Sloan Kettering Cancer Center, New York, NY, USA. [28]Department of Medicine, Washington University in Saint Louis, Saint Louis, MO, USA. [29]Division of Hematology and Oncology, Department of Medicine, University of California San Francisco, San Francisco, CA, USA. [30]Division of Clinical Hematology-Oncology, Barbara Ann Karmanos Cancer Institute, Detroit, MI, USA.

