## [Peer Review File · Nature Communications]

The Genetic Heterogeneity and Drug Resistance Mechanisms of Relapsed Refractory Multiple MyelomaReviewers' Comments:

Reviewer #1:

Remarks to the Author:

The Authors describe the results of the MMRF Molecular Profiling Initiative of RRMM with comprehensive genomic analyses of 511 of the 762 patients enrolled. They also compared mutation frequencies and types to previously characterized NDMM cases.

Some findings were expected including a higher incidence of classical tumor suppressors in RRMM and increased NR3C1 and CUL4B mutations in patients treated with corticoids and iMiDs, respectively.

However, the Authors also identify a number of new findings and characterize some the functional effects of these novel mutations, including in-frame alterations of TNFRSF17 and CD40. They also characterize the genome in uncommon MM cases that showed cryptic dysregulation of TNFRSF13/CD40/LTBR linking overexpression to translocations adjacent to strong plasma cell gene enhancers.

This is concise summary of many different genomic features of RR myeloma and is well written and presented.

However, a few disease correlations would be useful for the reader to interpret the diagnostic significance and/or clinical impact of these findings.

1. Since RAS pathway mutations are so common in RRMM, it would useful to present any correlations with length of disease, # of prior treatments or type of therapy for multiply mutated cases.
2. Were CD38 LOF mutations associated with loss of surface CD38 expression (or decreased CD38 RNA expression if flow data not available)?
3. Does mutation burden correlated with length of disease and/or # of therapies and/or MMID

Minor suggestions

P5,L127 (or P25-L706): Define briefly what is considered a mutation for calculation purposes. Are regulatory site and/or splice mutations included.

P5,L141: You state that MAF overexpression was likely due to MAF/A/B translocations. Was FISH correlative data not available for these cases?

Explicitly state if both RNA and DNA sequencing was performed on all cases and what criteria were used to fail a RNAseq analysis.

Cases with transcriptional upregulation of BCR-proximal associated kinases are not described. Was this pattern seen in the BCR-pathway mutated cases?

Reviewer #2:

Remarks to the Author:

In this manuscript currently considered for publication in Nature Communications, Vo et al reports targeted DNA sequencing and transcriptomic sequencing of over 500 patients with relapsed-refractory multiple myeloma enrolled in the MMRF molecular profiling protocol. Genetic abnormalities were detected in the NF-kB, the RAS-MAPK, MYC pathways as well as in proteins involved in DNA damage response and cell cycle regulation as well as RNA processing. In particular, based on their analysis, circa 60% of patients were noted to harbor mutations in NF-kB, a figure much larger than what previously reported. The authors also reports mutations in genes other than KRAS, NRAS and BRAF, that participate in RAS signaling and are mutated in a number of genetic syndromes. Finally, mutations that are presumably acquired secondary to selective pressure of therapies such as

IMiDs, steroids and CD38-targeting antibody daratumumab are also described.

Overall, the study appears technically well conducted, but of relatively little novelty compared to extensive literature in the field of myeloma genomics (see papers from Walker, Keats, Bolli, Maura, Landgren, etc..) that also includes papers exploring matched samples obtained from the same patient along their disease course as well as dedicated papers leveraging genomics to understand mechanisms of resistance to selected anti-MM drugs.

The major point in favor of this paper is the rather large sample size, however authors failed to take advantage of such a wealth of data to dive deep and perform functional studies to characterize the pathogenicity of newly reported mutated genes in MM (such as those associated with Rasopathies) or to support the claim that mutations detected in certain genes drive resistance to certain kind of therapies. The only functional study herein presented pertains to the activation of NFkB pathway.

Interestingly, the authors discuss genomic mutations arising as a consequence of certain treatment, such as IMiD, steroids and daratumumab, but do not touch upon any potential insight from genomics about mechanisms of resistance to proteasome inhibitors that are mainstay treatment of myeloma and to which patients reported herein must have been exposed and largely resistant and/or refractory.

Overall, my major concern is lack of novelty and impact in the field as data reported herein are confirmatory of other studies. The presence of functional validation of at least some of the data reported and hypothesis described would make this paper stand out as compared to others already published.

Reviewer #3:

Remarks to the Author:

This manuscript describes a study of Relapsed Refractory Multiple Myeloma (RRMM), particularly in the aspects of heterogeneity and drug resistance. This is a timely topic of wide interest to the Nature Communications readership. The investigators were able to analyze a study cohort of >500 cases (as part of an MMRF initiative) having deep tumor/normal sequence of a 1700-strong gene list, as well as expression data. They place their results in context against an almost 1000-member cohort from the CoMMpass study, ultimately characterizing the RRMM mutational landscape. Overall, this seems to be a thorough paper. Among its observations are the highly frequent pathway alterations in NF-kB and RAS pathways and the characterization of mutations in genes that confer resistance to front-line targeted therapies, including monoclonal antibodies.

I have a few criticisms that should be addressed.

1. I am not sure why the authors use Build 37 throughout, which has been obsolete for some time. It should use the current human reference.

2. Probabilistic thresholds are inconsistent, with at least 3 different usages reported: q-value of 0.1 (line 641), FDR of 0.05 (line 650), and a change of FDR to 0.2 for driver genes (line 675). Many analysts would consider the last value too high and it is unclear how that choice affects results. It would be good to treat this issue somewhat more rigorously, thus averting any unnecessary doubt about the results.

3. The cancer cell fraction (CCF) equation on lines 690-693 is similar, though not strictly identical to that in Dentre et al. Many readers will be confused with the product of CCF and multiplicity appearing on the left, rather than just CCF, given that multiplicity is determined by the second condition on line 693. In mathematics, "RHS" is a sometime abbreviation for "right hand side", though it is not clear that that is the meaning here, since the authors do not define it. Also, the authors invoke a CCF

threshold between clonal versus subclonal mutations of 0.8, without any justification or any examination of how sensitive their results are to this choice. As with FDR, it would be good to treat this issue somewhat more rigorously.

Point-by-Point Response to Reviewers' Comments
Manuscript NCOMMS-21-42577

Reviewer #1, expert in multiple myeloma genomics/clinical (Remarks to the Author):

The Authors describe the results of the MMRF Molecular Profiling Initiative of RRMM with comprehensive genomic analyses of 511 of the 762 patients enrolled. They also compared mutation frequencies and types to previously characterized NDMM cases.

Some findings were expected, including a higher incidence of classical tumor suppressors in RRMM and increased *NR3C1* and *CUL4B* mutations in patients treated with corticoids and iMiDs, respectively.

However, the Authors also identify a number of new findings and characterize some of the functional effects of these novel mutations, including in-frame alterations of *TNFRSF17* and *CD40*. They also characterize the genome in uncommon MM cases that showed cryptic dysregulation of *TNFRSF13/ CD40/LTBR* linking overexpression to translocations adjacent to strong plasma cell gene enhancers.

This is a concise summary of many different genomic features of RR myeloma and is well written and presented.

However, a few disease correlations would be useful for the reader to interpret the diagnostic significance and/or clinical impact of these findings.

RESPONSE: We are grateful for the reviewer's overall interest in our work and their insightful critiques. Through further data analysis and clarification of the text detailed below, we hope the reviewer agrees that we have addressed each of their concerns.

1. Since RAS pathway mutations are so common in RRMM, it would be useful to present any correlations with the length of disease, # of prior treatments, or type of therapy for multiply mutated cases.

RESPONSE: We thank the reviewer for this great suggestion. Unfortunately, the Multiple Myeloma Research Foundation did not collect detailed clinical data such as length of disease and treatment history in the Molecular Profiling clinical trial. We understand the value of this and plan to work with MMRF to retrospectively collect this outcome data for a follow-up analysis. Nevertheless, we agree that the high relevance of mutations in the RAS-MAPK pathway genes, including Rasopathy-associated genes, is striking in RRMM. We have revised the Discussion part to stimulate future studies to associate RAS-MAPK mutation status with clinical data:

"...Alterations in the RAS-MAPK pathway in MM are even more prevalent than alterations in the NF-κB pathway. In addition to the well-characterized *NRAS*, *KRAS*, and *BRAF* genes, our

study revealed that the germline Rasopathy genes represent a long tail of somatic alterations linking MM to these rare, congenital RAS-pathway-related diseases. If the designation of Rasopathy is extended to include mosaic conditions, such as keratinocytic epidermal nevus syndrome, alterations in *FGFR3* can also be integrated into this long tail, making alterations in the RAS-MAPK pathway even more prevalent in RRMM. Also, it is generally believed that the spectrum of mutations in the Rasopathies and in cancer minimally overlap, as exemplified by *PTPN11* and *BRAF*. One possible explanation is that cancer-associated RAS-MAPK mutations would be lethal for embryonic development, while the Rasopathy-associated RAS-MAPK mutations are too mild to evade apoptosis in malignant transformation. Interestingly, RAS-MAPK aberrations in RRMM are a conglomeration of both, making RRMM an ideal model to study strong and weak RAS activating events. Future studies should investigate the correlation between strong and weak RAS activators with the clinical history and outcome of RRMM patients..."

2. Were CD38 LOF mutations associated with loss of surface CD38 expression (or decreased CD38 RNA expression if flow data not available)?

RESPONSE: We thank the reviewer for this excellent question, which allows us to strengthen our argument. We plotted the gene expression profile of *CD38* in the RRMM cohort and highlighted cases with *CD38* mutations (missense, stop-gain, frameshift, and splice site). Patients with *CD38* mutations do associate with lower *CD38* expression than the rest of the cohort (Wilcoxon rank-sum test $P < 2 \times 10^{-3}$), which suggests that the biallelic inactivation of *CD38* play a role in drug resistance against monoclonal antibodies. We have added this figure to the revised manuscript as **Extended Data Fig. 9d**.

3. Does mutation burden correlate with length of disease and/or # of therapies and/or MMID?

RESPONSE: We thank the reviewer for this insightful question. Since clinical data for RRMM patients are not available, we cannot correlate mutation burden with the length of disease and the number of therapies within the RRMM cohort. However, as shown in **Extended Data Fig. 8a**, we compared mutation rate (the number of **point** mutations per captured megabases) in the RRMM patients relative to NDMM patients. Only point mutations with VAF $\geq 5\%$ were included for this analysis to account for the difference in sequencing depth between the two studies. The RRMM cohorts had a significantly higher mutation rate than the NDMM cohort, which provides indirect evidence that mutation burden correlates with the length of disease and the number of therapies.

Minor suggestions:

P5, L127 (or P25-L706): Define briefly what is considered a mutation for calculation purposes. Are regulatory sites and/or splice mutations included?

RESPONSE: We thank the reviewer for pointing out this detail. At the two instances mentioned by the reviewer, only point mutations (i.e. single-nucleotide variants, or SNVs) were considered. It is common practice in the field only to use SNVs for background mutation rate calculation. Functionally, SNVs can be synonymous, missense, stop-gain, start-loss, and splice-site. Regulatory site mutations were not included since our target capture panel does not generally cover non-coding regions. We have clarified this point in the revised main text and Methods.

P5, L141: You state that MAF overexpression was likely due to MAF/A/B translocations. Was FISH correlative data not available for these cases?

RESPONSE: We thank the reviewer for this question. FISH data were not routinely collected in the Molecular Profiling clinical trial. However, our results were confirmatory of Walker et al. (2015)¹ where the status of t(14;16) (*MAF*) and t(14;20) (*MAFB*) was validated thanks to tiled probe-set at the IGH, IGK, IGL loci. They also stated that "*APOBEC mutational signature is seen in 3.8% of cases and is linked to the translocation mediated deregulation of MAF and MAFB, a known poor prognostic factor*".

Explicitly state if both RNA and DNA sequencing was performed on all cases and what criteria were used to fail a RNA-seq analysis.

RESPONSE: We thank the reviewer for pointing out this detail.

- DNA sequencing was available for all cases. RNA sequencing was available for 510 out of 511 patients (as detailed in Supplementary Table 1). We have added this clarification

to the revised main text as followed:

"...RNA sequencing for **all but one** tumor samples was **available** and performed by capture transcriptome sequencing with a cohort average of 44.5M uniquely mapped reads..."

- Regarding quality control for RNA-seq data, we have added this information to the Methods in the revised main text:

"...Strand-specific RNA-seq FASTQ files were aligned to reference genome build hg19/GRCh37 in chimeric alignment mode by STAR aligner. **After alignment, libraries with ribosomal content $\geq 60\%$ mapped reads (i.e. failed ribosomal removal) and libraries with a low number of splice junctions ($<25^{\text{th}}$ percentile of all in-house libraries) were excluded from the final cohort.** Gene expression was quantified with *featureCounts*, and gene fusions were called using an in-house pipeline as previously described. Highly recurrent RNA chimeric transcripts (e.g., "trans-splicing") were filtered out from the reported fusions (Supplementary Table 4)...."

Cases with transcriptional upregulation of BCR-proximal associated kinases are not described. Was this pattern seen in the BCR-pathway mutated cases?

RESPONSE: We thank the reviewer for these suggestions. We believe the reviewer refers to the three kinases *LYN*, *SYK*, and *BTK*. We performed systemic outlier expression (Extended Data Fig. 6b), but our statistical framework (see Methods) did not nominate any RRMM case with *LYN*, *SYK*, and *BTK* as biological outliers (i.e. "upregulation"). In other words, the expression of *LYN*, *SYK*, and *BTK* have biological variability comparable to that of any expressed gene in our RRMM cohort, except those affected by structural rearrangement leading to overexpression.

Regarding how mutations in BCR-pathway may affect the expression of BCR-proximal associated kinases, we compare the expression of *LYN*, *SYK*, and *BTK* between cases with mutations in *CARD11*, *CD79B*, and *IRAK1* to the rest of the cohort. The differences in gene expression between the two groups did not reach any statistical significance for each gene (all Wilcoxon rank-sum test $P > 0.1$)

Reviewer #2, expert in multiple myeloma genomics/clinical (Remarks to the Author):

In this manuscript currently considered for publication in Nature Communications, Vo et al reports targeted DNA sequencing and transcriptomic sequencing of over 500 patients with relapsed-refractory multiple myeloma enrolled in the MMRF molecular profiling protocol. Genetic abnormalities were detected in the NF- κ B, RAS-MAPK, and MYC pathways as well as in proteins involved in DNA damage response, cell cycle regulation, and RNA processing. In

particular, based on their analysis, circa 60% of patients were noted to harbor mutations in NF- κ B, a figure much larger than what was previously reported. The authors also report mutations in genes other than *KRAS*, *NRAS*, and *BRAF*, that participate in RAS signaling and are mutated in a number of genetic syndromes. Finally, mutations that are presumably acquired secondary to selective pressure of therapies such as IMiDs, steroids and CD38-targeting antibody daratumumab are also described.

Overall, the study appears technically well conducted, but of relatively little novelty compared to extensive literature in the field of myeloma genomics (see papers from Walker, Keats, Bolli, Maura, Landgren, etc..) that also includes papers exploring matched samples obtained from the same patient along their disease course as well as dedicated papers leveraging genomics to understand mechanisms of resistance to selected anti-MM drugs.

RESPONSE: We thank the reviewer for their time and invaluable suggestions regarding our manuscript. We hope our additional data and clarification detailed below would further convince the reviewer and the editor about the novelty and impact of our study.

The major point in favor of this paper is the rather large sample size, however authors failed to take advantage of such a wealth of data to dive deep and perform functional studies to characterize the pathogenicity of newly reported mutated genes in MM (such as those associated with Rasopathies) or to support the claim that mutations detected in certain genes drive resistance to certain kinds of therapies. The only functional study herein presented pertains to the activation of NF- κ B pathway.

RESPONSE: We thank the reviewer for recognizing one of our strengths. Indeed, this would be the largest clinical sequencing study in relapsed refractory multiple myeloma to date. Together with our effort to reanalyze almost 1,000 newly diagnosed MM cases in the CoMMpass study, our study is probably the most thorough exploration of the myeloma molecular genetic landscape. We would like to emphasize that a large portion of clinical sequencing papers, including high-profile ones, do not usually include functional data. Functional characterization of selected alterations merits a separate follow-up study so that experimental results and discussions can be fleshed out in details. In the initial submission, we went beyond our duties to characterize novel intriguing inframe indels in *BCMA* and *CD40* functionally. Given that we have uncovered a wide spectrum of alterations in the NF- κ B, RAS-MAPK (including Rasopathy-associated ones), *MYC* pathways, and drug resistance mechanism, it is not optimal to present additional experimental results due to space constraints.

Nevertheless, we took the reviewer's recommendations and performed functional studies of our newly identified *IL6ST* alterations in MM (**Fig. 3f**). It has been reported that interleukin 6 cytokine family signal transducer (*IL6ST* or gp130) could activate the RAS-MAPK pathway through its association with PTPN11, as well in the JAK/STAT pathway through JAKs^{2,3}. The pattern of mutations in *IL6ST* in our RRMM cohort was strikingly similar to those described in inflammatory hepatocellular carcinoma (IHCA)^{4,5}. In-frame indels and recurrent point substitutions affected the D2 domain of *IL6ST* (**Fig. 3e**), which could facilitate its dimerization

even in the absence of IL-6^{4,5}. While IHCA-associated *IL6ST* variants almost always cluster from codon 168 to 216, we observed mutations that appeared earlier in the D2 domain (V136E, E138K) and far later in the D3 domain (K303T, D312_S314dup) (**Fig. 3e**). These rare mutants could mediate STAT3 activation as robustly as the more common ones (**Fig. 3f**). In addition, *IL6ST* variants were significantly enriched in RRMM compared to NDMM ($P < 0.001$, Fisher's exact test), which reflects the progressive independence of the myeloma cells from bone marrow cytokines in some advanced patients.

Interestingly, the authors discuss genomic mutations arising as a consequence of certain treatment, such as IMiD, steroids, and daratumumab, but do not touch upon any potential insight from genomics about mechanisms of resistance to proteasome inhibitors that are the mainstay treatment of myeloma and to which patients reported herein must have been exposed and largely resistant and/or refractory.

RESPONSE: We thank the reviewers for this excellent comment. We are also intrigued by the lack of an apparent mechanism for the resistance against the proteasome inhibitor drug class, despite our unbiased effort to search for the enrichment of mutations and copy number alterations in RRMM compared to NDMM (**Fig. 4a-b, Extended Data Fig. 8a-c**). We hope that future studies using genome-wide CRISPR-screening techniques could help us narrow down the candidates. We have added this point to the Discussion as a limitation of our study.

Overall, my major concern is lack of novelty and impact in the field as data reported herein are confirmatory of other studies. The presence of functional validation of at least some of the data reported and hypothesis described would make this paper stand out as compared to others already published.

RESPONSE: We thank the reviewer for their genuine suggestions to improve the quality of our

manuscript. As detailed above and in the revised main text, we have performed additional functional studies on *IL6ST* mutations that have never been reported in multiple myeloma literature. This important finding highlights that a subset of RRMM patients might receive clinical benefits from existing approved JAK/STAT inhibitors.

However, we would still like to underline other novelties and impacts of our studies, which might not have come across clearly in the initial submission:

- Our findings of alterations in the NF- κ B pathway are more than just confirmatory of previous publications. For example, inframe deletions of *CD40*, inframe insertion of *BCMA*, and 5' deletions of *MAP3K14* already existed in the CoMMpass dataset (Fig. 2c,d,e, cases in deep purple with prefix MMRF) but have never been reported. We were able to identify these thanks to our comprehensive bioinformatics pipeline that could call mid-range to longer indels. We were also very thorough in interpreting genetic results. While mutations in *IRAK1* might have been reported elsewhere, we were the first to perform multiple sequence alignment to reveal that mutations in *IRAK1* are highly recurrent and render the protein kinase-dead. This finding could shed more light on this intriguing NF- κ B and apoptosis regulator.
- Unlike other landscape studies, where alterations in the RAS-MAPK were simply tabulated, we took an unusual deep dive and inspected all variants, including those at the long tails. With scrupulous curation, we could identify alterations associated with the Rasopathies. This finding presents a substantial conceptual advance in the basic biology of RAS-MAPK and of multiple myeloma. As discussed in the revised Discussion, RRMM could now be viewed as an ideal model for both weak and strong RAS-MAPK activators. Our finding could also serve as a starting point for future clinical to investigate any differences in clinical outcomes and drug responses between strong and weak RAS-MAPK activators.
- Our discover of alterations in *CD38* is non-trivial and could have clinical impact. While the incidence of *CD38* alterations was relatively low (3%), two patients already harbored distinct mutations that converged into the inframe exon skipping affected codon 221 to 250, presumably only disrupting the epitope of daratumumab (**Fig.4i-j**) while retaining a major portion of the extracellular domain. In theory, such patients could still benefit from isatuximab, another monoclonal therapy targeting *CD38*. As Lee et al pointed out⁶, the epitope of isatuximab is composed of residues from codons 34 to 189, thus completely unaffected by this exon skipping. Future structural and clinical studies should explore this direction to widen the therapeutic options for relapsed patients affected by this recurrent alteration.

We have extended our Discussion to include the above points.

Reviewer #3, expert in clonal evolution and MM genomics (Remarks to the Author):

This manuscript describes a study of Relapsed Refractory Multiple Myeloma (RRMM), particularly in the aspects of heterogeneity and drug resistance. This is a timely topic of wide interest to the Nature Communications readership. The investigators were able to analyze a study cohort of >500 cases (as part of an MMRF initiative) having deep tumor/normal sequence of a 1700-strong gene list, as well as expression data. They place their results in context against an almost 1000-member cohort from the CoMMpass study, ultimately characterizing the RRMM mutational landscape. Overall, this seems to be a thorough paper. Among its observations are the highly frequent pathway alterations in NF- κ B and RAS pathways and the characterization of mutations in genes that confer resistance to front-line targeted therapies, including monoclonal antibodies.

RESPONSE: We thank the reviewer for the enthusiasm in our manuscript. We are also grateful for their constructive critiques. We hope that by addressing the concerns, we have significantly improved the persuasiveness of our arguments and made the manuscript suitable for publication.

I have a few criticisms that should be addressed.

1. I am not sure why the authors use Build 37 throughout, which has been obsolete for some time. It should use the current human reference.

RESPONSE: We thank the reviewer for this question. The bioinformatics results in this study were generated using the GRCh37/hg19 reference genome as part of the integrative sequencing protocols in our Clinical Laboratory Improvement Amendments (CLIA)-certified sequencing laboratory at the University of Michigan. The results have also been reviewed and approved for sending out to physicians at institutions participating in the Molecular Profiling and the MyDRUG clinical studies. In addition, GRCh37/hg19 remains the standard build for clinical labs, with more than 85% of labs participating in CAP proficiency testing currently using this build.

However, we agree with the reviewer that the current human reference genome (hg38) contains some improvements over the older build. Some readers may be interested in viewing the results in hg38. Therefore, we have used the tool *liftOver* to provide coordinates in hg38 for somatic mutations (revised Supplementary Table 2), copy number segments (revised Supplementary Table 3), and gene fusions (revised Supplementary Table 4).

2. Probabilistic thresholds are inconsistent, with at least 3 different usages reported: q-value of 0.1 (line 641), FDR of 0.05 (line 650), and a change of FDR to 0.2 for driver genes (line 675). Many analysts would consider the last value too high and it is unclear how that choice affects results. It would be good to treat this issue somewhat more rigorously, thus averting any unnecessary doubt about the results.

RESPONSE: We thank the reviewer for these very important suggestions.

- Our choice of the q-value threshold of 0.05 for APOBEC signature enrichment followed

Roberts et al.⁷ who first introduced this analysis. Papers since then have also adapted this threshold^{8,9}.

- For consistency, we updated the threshold of GISTIC2.0 analysis from $q < 0.1$ to $q < 0.05$, thus making a minor revision to Fig. 1b (the dashed green line). Since the annotated gain and loss peaks (on top of Fig. 1b) have q-values much smaller than 0.05, this change does not affect our results.
- Regarding the False Discovery Rate (FDR) threshold for cancer driver identification, we agree with the reviewer that $FDR < 0.2$ might have appeared too "forgiving" at first. However, this choice of FDR threshold reflected our efforts to balance the statistical sense and the biological sense of the analysis.

First, our consensus approach was inspired by the landmark paper by Bailey et al.¹⁰ Using a catalog of 26 statistical tools, the author devised a strategy to compile the results and nominate cancer drivers in TCGA data. Each statistical tool employed different criteria to define what is considered significant, such as the clustering of mutation in protein sequence (*OncodriveCLUST*), accumulation of mutations that have higher predicted functional impact (*OncodriveFML*), enrichment of mutation based on inferred background mutation processes (*MutSigCV* and *MutSig2CV*), machine learning (*20/20+*), and so on. Predictably, the tools' outputs widely diverged, and even their sophisticated composite score still missed some known drivers. Therefore, Bailey et al. cautiously highlighted the need for literature review and expert curation when discovering candidate cancer drivers. For example, given its well-known tumor-suppressing function and mutation frequency at 18%, there is no doubt that *TP53* is a driver in multiple myeloma. Yet, it was associated with a surprisingly high q-value at 0.13 by *OncodriveCLUST* in our cohort. A strict FDR cut-off at 0.05 or 0.1 would have missed this call. Likewise, the important gene *EP300* would have barely made it with *MutSigCV* with a q-value of 0.16. Interestingly, even Bailey et al. used a wide range of q-value cutoff for their inclusion criteria, such as 0.0001, 0.01, 0.05, 0.1, and even 0.25 without further justification (please see the **STAR Methods** in their *Cell* paper). It is likely that they also recognized the problem we were facing, that each tool requires a distinct threshold for the appropriate interpretation of the results.

Second, it's worth mentioning that the tools we used were originally developed for whole-genome sequencing and whole-exome sequencing data. The background of the statistical test is 20,000 coding genes. While we have tried our very best to adapt each tool for our 1,700-gene panel, they might not have been fully optimized. We suspected that if our data had been whole-exome sequencing, the output p-values and q-values would have been much smaller.

Eventually, we opted for an FDR cut-off at 0.2 to accommodate all tools, followed by "voting" (at least two out of five) to nominate drivers. This approach can be viewed as a simplified version of the weighted composite score proposed by Bailey et al. In retrospect, not only did our final list of candidates agree with known drivers in MM

literature^{11,12}, but it also revealed new ones associated with drug resistance in RRMM (*CRBN*, *CULAB*, and *NR3C1*).

3. The cancer cell fraction (CCF) equation on lines 6901-693 is similar, though not strictly identical to that in Dentre et al. Many readers will be confused with the product of CCF and multiplicity appearing on the left, rather than just CCF, given that multiplicity is determined by the second condition on line 693. In mathematics, "RHS" is a sometime abbreviation for "right hand side", though it is not clear that that is the meaning here, since the authors do not define it. Also, the authors invoke a CCF threshold between clonal versus subclonal mutations of 0.8, without any justification or any examination of how sensitive their results are to this choice. As with FDR, it would be good to treat this issue somewhat more rigorously.

RESPONSE:

We thank the reviewer for these very important suggestions. While the formulas we used to calculate CCF are essentially the same as those from Dentre et al., we agree that the way we presented it in the initial submission could have confused the readers. We thus revised this part in the **Methods** as the following:

The cancer cell fraction (CCF) of a variant (including point mutation or small indel) i was defined as in Dentre et al. Briefly, the relationship between mutation multiplicity m_i of a variant and its cancer cell fraction CCF_i is considered as the following:

$$u_i = CCF_i m_i$$

Where:

$$u_i = \frac{(1 - purity) * 2 + purity * Local_Copy_Number}{purity} * VAF_i$$

Ideally, a clonal mutation should have a CCF of 1.0 (100% of tumor cells should contain this mutation), and a subclonal mutation should have a CCF less than 1.0. Therefore, the multiplicity m_i can be calculated as:

$$m_i = \begin{cases} u_i, & \text{if } u_i \geq 1 \\ 1 & \text{if } u_i < 1 \end{cases}$$

Regarding the clonality definition, it is a common practice in the field to lower the threshold to 0.8-0.9 rather than using the literal definition of $CCF = 1.0$ for clonal mutation. This lower

threshold could account for uncertainties in estimating local copy number and tumor purity. We followed the precedences set by Rasche et al., Kamran et al., and Zhang et al.^{13,14,15} Nevertheless, whether the threshold is 1.0 or 0.8, we would like to assure the reviewer that it does not affect the main point of Fig.3a, that "*clonal RAS G12, G13, Q61, and BRAF V600E were strictly mutually exclusive with each other*".

Reference

1. Walker, B. A. *et al.* APOBEC family mutational signatures are associated with poor prognosis translocations in multiple myeloma. *Nat. Commun.* 6, 6997 (2015)
2. Ohtani, T. *et al.* Dissection of signaling cascades through gp130 in vivo: reciprocal roles for STAT3- and SHP2-mediated signals in immune responses. *Immunity* 12, 95–105 (2000)
3. Fukada, T. *et al.* Two Signals Are Necessary for Cell Proliferation Induced by a Cytokine Receptor gp130: Involvement of STAT3 in Anti-Apoptosis. *Immunity* vol. 5 449–460 (1996)
4. Rebouissou, S. *et al.* Frequent in-frame somatic deletions activate gp130 in inflammatory hepatocellular tumours. *Nature* 457, 200–204 (2009)
5. Poussin, K. *et al.* Biochemical and functional analyses of gp130 mutants unveil JAK1 as a novel therapeutic target in human inflammatory hepatocellular adenoma. *Oncoimmunology* 2, e27090 (2013)
6. Lee, H. T. *et al.* Crystal structure of CD38 in complex with daratumumab, a first-in-class anti-CD38 antibody drug for treating multiple myeloma. *Biochem. Biophys. Res. Commun.* 536, 26–31 (2021)
7. Roberts, S. A. *et al.* An APOBEC cytidine deaminase mutagenesis pattern is widespread in human cancers. *Nat. Genet.* 45, 970–976 (2013)
8. Cancer Genome Atlas Research Network *et al.* Integrated genomic and molecular characterization of cervical cancer. *Nature* 543, 378–384 (2017)
9. Glaser, A. P. *et al.* APOBEC-mediated mutagenesis in urothelial carcinoma is associated with improved survival, mutations in DNA damage response genes, and immune response. *Oncotarget* vol. 9 4537–4548 (2018)
10. Bailey, M. H. *et al.* Comprehensive Characterization of Cancer Driver Genes and Mutations. *Cell* 174, 1034–1035 (2018)
11. Walker, B. A. *et al.* Identification of novel mutational drivers reveals oncogene dependencies in multiple myeloma. *Blood* 132, 587–597 (2018)
12. Maura, F. *et al.* Genomic landscape and chronological reconstruction of driver events in multiple myeloma. *Nat. Commun.* 10, 3835 (2019)
13. Rasche, L. *et al.* Spatial genomic heterogeneity in multiple myeloma revealed by multi-region sequencing. *Nat. Commun.* 8, 268 (2017)
14. Kamran, S. C. *et al.* Integrative Molecular Characterization of Resistance to Neoadjuvant Chemoradiation in Rectal Cancer. *Clin. Cancer Res.* 25, 5561–5571 (2019)
15. Zhang, H. *et al.* Sex difference of mutation clonality in diffuse glioma evolution. *Neuro-Oncology* vol. 21 201–213 (2019)

Reviewers' Comments:

Reviewer #1:

Remarks to the Author:

Appreciate the Authors responses to my and the other reviewers' comments.

In response, the Authors added a few pieces of correlative data (with CD38 levels correlating with splice mutations) and functional data (STAT activation demonstrated in cases with certain IL6ST D2 and D3 domain mutations) which supports the significance of the observed genomic findings in those cases.

Reviewer #2:

Remarks to the Author:

The authors have addressed most critiques of the 3 reviewers and conduct and thorough and careful resubmission. I do not have any further critique and deem this revised manuscript acceptable for publication.

Reviewer #3:

Remarks to the Author:

Authors did a good job with revision. No additional comments.

REVIEWERS' COMMENTS

Reviewer #1 (Remarks to the Author):

Appreciate the Authors responses to my and the other reviewers' comments.

In response, the Authors added a few pieces of correlative data (with CD38 levels correlating with splice mutations) and functional data (STAT activation demonstrated in cases with certain IL6ST D2 and D3 domain mutations) which supports the significance of the observed genomic findings in those cases.

RESPONSE: We thank the reviewer for the accurate summary of our additional data. We would like to extend our gratitude to the reviewer for their very insightful questions and suggestions in the first round of revision.

Reviewer #2 (Remarks to the Author):

The authors have addressed most critiques of the 3 reviewers and conduct and thorough and careful resubmission. I do not have any further critique and deem this revised manuscript acceptable for publication.

RESPONSE: We thank the reviewer for the approval of our revision. We would like to extend our gratitude to the reviewer for their constructive critiques and suggestions in the first round of revision.

Reviewer #3 (Remarks to the Author):

Authors did a good job with revision. No additional comments.

RESPONSE: We thank the reviewer for the endorsement of our revision. We would like to extend our gratitude to the reviewer for their great suggestions and enthusiasm in the first round of revision.